# Correlation analysis of mitochondrial DNA maintenance-related genes with HCC prognosis, tumor mutation burden and tumor microenvironment features

Fan Rong[1☯], Bin Cheng[1☯], Ling Guo[1], Shaobo Zeng[2], Xunliang Xu[3], Zhongji Meng[1]*

**1** Department of Infectious Diseases, Institute of Biomedical Research, Regulatory Mechanism and Targeted Therapy for Liver Cancer Shiyan Key Laboratory, Hubei provincial Clinical Research Center for Precise Diagnosis and Treatment of Liver Cancer, Taihe Hospital, Hubei University of Medicine, Shiyan, China, **2** Department of Hepatobiliary Pancreatic Surgery, Wudangshan Branch of Taihe Hospital, Hubei University of Medicine, Shiyan, China, **3** Hubei provincial Clinical Research Center for Precise Diagnosis and Treatment of Liver Cancer, Taihe Hospital, Hubei University of Medicine, Shiyan, China

☯ These authors contributed equally to this work.
* zhongjimeng@163.com

## Abstract

### Background

Mitochondrial DNA (mtDNA) is an important genetic material in eukaryotic cells. Mitochondrial DNA maintenance-related gene (mtDNA MRG) variants contribute to mitochondrial dysfunction in cancer progression and are associated with cancer prognosis. However, the mechanism of mtDNA MRGs in the tumor microenvironment (TME) of hepatocellular carcinoma (HCC) remains unclear.

### Methods

Data for a total of 487 HCC samples were collected from The Cancer Genome Atlas (TCGA) and the Gene Expression Omnibus (GEO). The mitochondrial regulatory pathway gene set was downloaded, and 22 mtDNA MRGs were identified by screening. Based on these 22 genes, the HCC samples were grouped by unsupervised clustering based on a machine learning model. Principal component analysis (PCA) was used to construct the mtDNA score model, and the relationships between the mtDNA score and clinicopathological features, tumor mutation burden (TMB), TME cell infiltration and biological processes were analyzed.

### Results

The expression of 22 mtDNA MRGs significantly different in HCC samples *vs.* normal controls. In this study, HCC samples were divided into three molecular subtypes based on the expression of mtDNA MRGs. The three subtypes exhibit different

**Data availability statement:** The HCC RNA transcriptome sequencing data and clinical information were obtained from The Cancer Genome Atlas (TCGA) (http://cancergenome. nih.gov/) and Gene Expression Omnibus (GEO) (https://www.ncbi.nlm.nih.gov/geo/). TCGA and GEO databases are public databases where relevant data can be downloaded directly.We have already upload figures to figshare, DOI: 10.6084/m9.figshare.27089542.

**Funding:** This work was supported by the Hubei Provincial Technology Innovation Project (2023BCB129); the Project of Creative Research Groups of Hubei Province (2023AFA023); Special Fund for Artificial Liver of the Beijing Hepatobiliary Charity Foundation (RGGJJ-2021-026); Ligan project of the Beijing Hepatobiliary Charity Foundation (iGandanF-1082024-LG001).

**Competing interests:** The authors have declared that no competing interests exist.

clinical characteristics and immune infiltration profiles, and the cell infiltration profiles corresponded to the immune rejection, immune inflammation, and immune-desert phenotypes, respectively. A total of 740 core genes were obtained from different molecular subtypes, and these genes were divided into three gene subtypes. The mtDNA score model, which can be used to assess tumor immune cell invasion, clinicopathological features, genetic variation, and prognosis, was subsequently constructed. A high mtDNA score was associated with a high mutation burden, high clinical stage and poor prognosis.

## Conclusions

mtDNA MRGs play important roles in HCC TMB, prognosis, clinicopathological features and the immune microenvironment. The mtDNA score can be used to evaluate HCC prognosis, TMB and the immune microenvironment, thereby providing guidance for treatment decision making and prognosis prediction in HCC patients.

## Introduction

Hepatocellular carcinoma (HCC) is the most common type of primary liver cancer, accounting for approximately 90% of all primary liver cancers [1]. It is also one of the top four causes of cancer-related death worldwide; 700,000 new cases are diagnosed worldwide each year, half of which are in China [2]. Liver transplantation and early surgical resection can improve the prognosis of HCC [3]. However, because the early symptoms of HCC patients are not typical, most HCC patients are often diagnosed at a late stage, and the mortality rate of HCC patients is still unusually high [4]. Therefore, there is an urgent need to elucidate the pathogenesis of HCC, identify potential biomarkers for early diagnosis, and develop novel therapies for HCC patients.

Mitochondria are ubiquitous in almost all eukaryotic cells and are mainly involved in oxidative phosphorylation (OXPHOS), reactive oxygen species (ROS) production, the tricarboxylic acid (TCA) cycle, heme synthesis, amino acid metabolism, apoptosis regulation, innate immunity, calcium homeostasis and other important processes [5,6]. mtDNA is a double-stranded closed circular DNA molecule composed of 16,569 base pairs that encodes 13 peptides in humans, as well as 22 tRNAs and 2 rRNAs required for mitochondrial protein synthesis [7,8]. mtDNA replication, repair, modification and stability maintenance play important roles in the maintenance of mtDNA homeostasis, aiming to maintain the integrity of mitochondrial structure and function [9]. Genes involved in mtDNA homeostasis maintenance are collectively referred to as mtDNA maintenance-related genes (mtDNA MRGs). Recent studies have shown that mtDNA MRG variants cause mitochondrial defects and dysfunction and lead to various mitochondrial diseases [10]. Mitochondrial dysfunction leads to OXPHOS disruption and increases ROS production, which is involved in HCC progression [11]. Moreover, mitochondrial dysfunction is associated with tumor chemotherapy resistance [12].

The tumor microenvironment (TME) also plays a considerable role in tumor cell progression [13], regulating tumor cell proliferation, invasion, immune escape and the response to immunotherapy through various signaling pathways [14]. The characteristics of cell infiltration in the TME can predict the response to immune checkpoint blockade (ICB) therapy [15]. A comprehensive analysis of the heterogeneity and complexity of the TME landscape may reveal different tumor immunophenotypes, provide an important basis for guiding treatment and predicting the immunotherapy response, and help to identify new therapeutic targets.

mtDNA MRGs have been identified as participants in cancer occurrence and progression [16]. For example, mitochondrial ND3, ND4, and ND5 mutation-mediated ROS elevation is a key factor in cancer pathogenesis and the generation of cancer-promoting phenotypes [17,18]. Increasing evidence shows that mitochondrial dysfunction is closely related to the occurrence and progression of HCC. High expression of APEX1 is associated with resistance to sorafenib and anti-programmed death 1 (PD-1) therapy in HCC patients [19]. The expression level of OGG1 in HCC patients is higher than that in healthy people. OGG1 promotes the occurrence of liver cancer by promoting the expression of cell cycle-related proteins in tumor cells and the repair of oxidative DNA damage [20]. SSBP1 is a key protein in mtDNA replication and is involved in tissue-specific mtDNA wasting diseases. Most of these studies focused on a single mtDNA gene [21]. However, the complex interactions among mtDNA MRGs in the development and progression of HCC are unclear.

In this study, we focused on exploring the important role of mtDNA MRGs in the HCC TME, constructed a mtDNA score model, and evaluated its value in predicting HCC prognosis, TMB, immune microenvironment features, and immunotherapy response. This study may provide more information on the molecular mechanism and prognostic biomarkers of HCC.

## Materials and methods

### Data acquisition and preprocessing

The HCC RNA transcriptome sequencing data and clinical information were obtained from The Cancer Genome Atlas (TCGA) (http://cancergenome.nih.gov/) and Gene Expression Omnibus (GEO) (https://www.ncbi.nlm.nih.gov/geo/). Data were obtained for 374 HCC samples and 50 normal samples from the TCGA on June 17, 2023, and 115 primary HCC tissues and 52 adjacent tissues from the GEO (GSE76427) cohort [22]. TCGA RNA sequencing data (FPKM format) were downloaded from the Genomic Data Commons (GDC, https://portal.gdc.cancer.gov/) and then used the fpkm function of the "limma" package in R to convert the FPKM value of the RNA data to the TPM value. For the GSE76427 data from the Illumina platform, the normalized matrix file was directly downloaded [23]. Patients without survival information were excluded. The list of mtDNA MRGs was obtained from the MitoCarta3.0database(https://www.broadinstitute.org/mitocarta/mitocarta30-inventory-mammalian-mitochondrial-proteins-and-pathways), and then, the mtDNA MRGs were obtained for integration and batch correction, the performance of oPLS-DA model was assessed by R²Y (goodness of fit) and Q² (goodness of prediction) through cross-validation, along with permutation testing to evaluate robustness. Data were downloaded from the publicly available database hence it was not applicable for additional ethical approval.

### Unsupervised clustering of mtDNA MRGs

A total of 22 mtDNA MRGs were identified in this study and are listed as follows: mtDNA replication (DNA2, EXOG, LIG3, MGME1, POLG, POLG2, POLRMT, RNASEH1, SSBP1, TFAM, TFB2M, TOP1MT, TOP3A, TWNK), mtDNA repair (APEX1, EXOG, LIG3, OGG1, POLB, POLG, PRIMPOL, RECQL4, UNG), mtDNA modifications (METTL4), mtDNA stability and decay (ENDOG, EXOG, MGME1). Based on the mtDNA MRGs expression levels, HCC samples were classified for further analysis by using the "ConsensusClusterPlus" package in R for unsupervised cluster analysis, and 50 repetitions were performed with pItem = 0.8, pFeature = 1 to verify the stability of the subtypes. The number of cluster k-values were increased from 2 to 9, the k-values with better clustering stability were selected according to the clustering effect [24].Kaplan–Meier curves were used to assess the overall time to survival (OS) of patients with different clusters.

## Gene set variation analysis (GSVA) and functional annotation

GSVA is a non-parametric, unsupervised method that is mainly used to estimate pathway changes in experimental data-sets and differences in activity during biological processes [25]. To further explore the biological functions and signaling mechanisms between mtDNA MRG clusters, "GSVA" in the R package was used for GSVA enrichment analysis. "c2.cp.kegg.v7.2.symbols.gmt" was downloaded from the MSigDB database (http://www.gsea-msigdb.org/gsea/msigdb). The p value was adjusted according to the false discovery rate (FDR), with adjusted P < 0.05 as the cut-off criterion.

## Analysis of immune cell differences in the TME

To elucidate the intricate mechanisms underlying tumor-immune cell interactions, Pornpimol Charoentong, et al characterized the intratumoral immune landscapes and the cancer antigenomes from 20 solid cancers and created The Cancer Immunome Atlas (https://tcia.at/). Using machine learning, the determinants of tumor immunogenicity were identified and a quantitative scoring scheme called immunophenotypic scoring was developed. The gene set of TME-infiltrating immune cells obtained from Pornpimol Charoentong's study included activated B cells, activated CD4 T cells, natural killer T cells and regulatory T cells [26]. The single sample gene set enrichment analysis (ssGSEA) algorithm was used to quantify the relative abundance of individual infiltrating cells in the HCC TME. The relative abundance of each immune cell type was represented by an enrichment score in ssGSEA analysis and normalized to unity distribution from 0 to 1 [27]. The degree of infiltration of immune cells in each sample was indicated by the enrichment fraction calculated via ssGSEA.

## Identification of mtDNA MRG gene clusters

According to the consensus clustering algorithm, 22 mtDNA MRGs were divided into three different subgroups. Based on the above three subgroups of mtDNA MRGs, the R package "limma" was used to screen differentially expressed genes (DEGs), and an adjusted p value less than 0.001 was used as the screening criterion. The Venn diagram package was used to identify overlapping genes between different subgroups, and KEGG enrichment analysis was subsequently performed to explore the potential related biological functions and biological pathways.

Consensus ClusterPlus cluster analysis in R was used to identify three gene clusters, the Kaplan–Meier method was used to analyze the prognosis for groups enriched in the three gene clusters, and the log-rank test was used to evaluate the statistical significance. The "pheatmap" package was used to create a heatmap showing the relationships between gene clusters and clinicopathological features.

## mtDNA score

To quantify the characteristics of mtDNA MRGs in individual HCC patients, a scoring system called the mtDNA score was constructed using principal component analysis (PCA). Combined with previous studies, core genes were extracted from DEGs, and a univariate Cox regression model was used to assess prognosis according to each gene in the feature model. Then, PCA was performed to extract principal components, and principal components 1 and 2 were selected to construct the signature score model. This approach focuses on the scores of the set with the largest number of well-correlated (or inversely correlated) gene blocks in the set, while eliminating as much as possible the contributions of genes that do not track other members [27]. We adopted a formula similar to previous studies to define the mtDNA score [28]:

$$mtDNA\ score = \sum(PC1i + PC2i)$$

i represents the expression of mtDNA phenotype-associated genes. Patients were divided into high and low mtDNA score groups based on maximum choice ranking statistics.

### Correlation of the mtDNA score with the TMB, clinicopathological features, and immunotherapy response status

According to the mtDNA score, the samples were divided into a high mtDNA score group and a low mtDNA score group. A waterfall map was drawn using the "maftools" package to show the difference in mutation rate between the high-mtDNA score group and low-mtDNA score group. Differences in the mtDNA score according to pathological stage and survival status were analyzed. The percentage chart was drawn using the "plyr" package, and the box plot was drawn using the "ggpub" package. The "limma" package was used to compare mtDNA score differences between different mtDNA MRG cluster subgroups and gene subgroups. The immune checkpoint inhibitor (ICI) immunophenotype score (IPS) was downloaded from the cancer immune group database, and differences in the response to CTLA-4 and PD-1 blockade immunotherapies between groups were analyzed.

### Statistical analysis

Spearman's and distance correlation analyses were used to determine the correlation coefficient between TME infiltration of immune cells and mtDNA MRG expression. One-way ANOVA and the Kruskal–Wallis test were used to analyze differences between three or more groups. Survival curves for prognostic analyses were generated using the Kaplan–Meier method, and logarithmic rank tests were used to determine the significance of differences. All the statistical analyses were performed using R version 4.1.0. $P < 0.05$ was considered to indicate statistical significance.

## Results

### Genetic variation and transcriptional overview of mtDNA MRGs in HCC

In this study, to ensure data reliability, we first performed batch correction on the dataset (S1A Fig and B). The orthogonal partial least squares-discriminant analysis (oPLS-DA) model was then applied to assess residual batch effects (S1C Fig, D). The oPLS-DA model demonstrated high efficacy, with an R²Y of 0.767, indicating that 76.7% of the sample classification variance could be explained by the model. The cross-validation predictive ability (Q²Y = 0.672, threshold >0.5) confirmed the model's robustness (P < 0.05, permutation test), excluding the possibility of random classification. These results suggest that the model is both statistically reliable (high R²Y) and predictively valid (high Q²Y). To further validate the model's performance, we conducted a permutation test (100 random label shuffles) (S1D Fig). The real model's R²Y/Q²Y (red line) significantly outperformed the random distribution (gray bars, P = 0.05), confirming that the observed classification was not due to chance. This stringent validation ensures that the identified differentially expressed genes and pathways are biologically meaningful rather than artifacts.

The levels of transcripts and genetic variants of 22 mtDNA MRGs were investigated, and the results revealed that the expression of 22 mtDNA MRGs was upregulated in HCC tissues compared with healthy tissues (Fig 1A). To further understand the genetic variation of mtDNA MRGs in HCC, the copy number variation (CNV) and somatic cell mutation rate of mtDNA MRGs in HCC samples were summarized. Among the 371 HCC samples, 28 (7.55%) had mutations in mtDNA MRGs. The major mutated genes were RECQL4 (2%), APEX1 (1%), DNA2 (1%), LIG3 (1%), POLG (1%), POLG2 (1%), POLRMT (1%), and TOP1MT (1%) (Fig 1B). Among them, 18 genes were significantly related to prognosis (Fig 1C-T). According to the above analysis, the genetic mutations and expression changes of mtDNA MRGs were highly heterogeneous between normal samples and HCC samples, suggesting that the imbalance in the expression of mtDNA MRGs plays an important role in the occurrence and development of HCC.

### Identification of molecular subtypes of mtDNA MRGs

The interactions and correlations among mtDNA MRGs were demonstrated through a network interaction diagram, as shown in (Fig 2A). POLG and ENDOG were found to be protective factors in terms of the prognosis of HCC. K = 3 was the best cluster value for grouping HCC patients according to the expression of the mtDNA MRGs; the clusters were named as

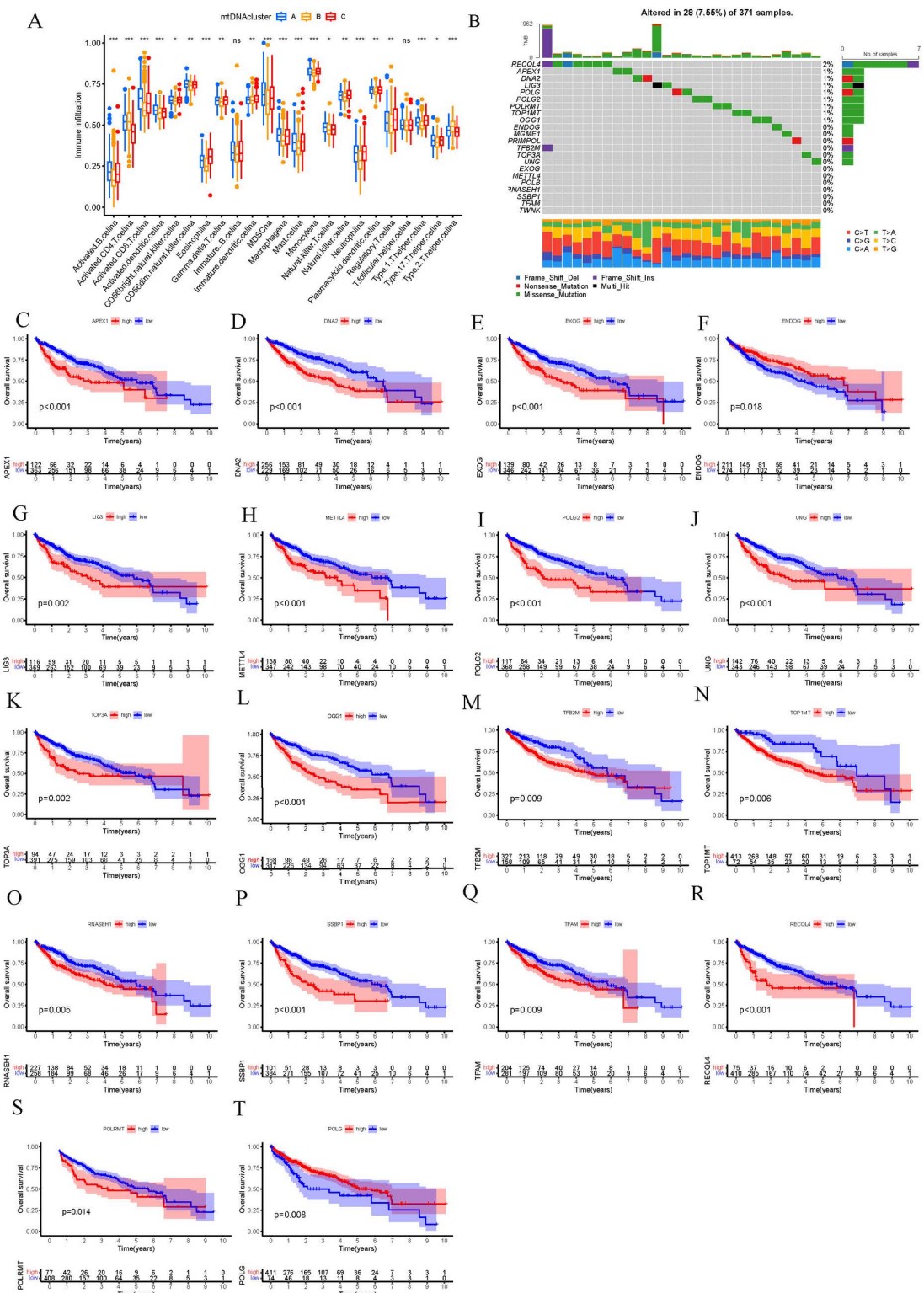

**Fig 1. Overview of mtDNA MRG mutations and transcription in HCC.** A. Expression of mtDNA MRGs in healthy tissue and HCC tissue. B. mtDNA MRG mutation frequencies in 371 HCC patients. The histogram above shows the total TMB. The numbers on the right indicate the frequency of mutations in each gene. The bar chart on the right shows the proportion of each mutation. The stacked bar chart below shows the converted scores. C-T. Survival analysis

of HCC patients according to mtDNA MRG expression (P<0.05). APEX1, Apurinic/apyrimidinic endonuclease 1; DNA2, ATP-dependent helicase/nuclease 2; EXOG, Endo/exonuclease G-like; EXDOG, endonuclease G; LIG3, ligase III; METTL4, Methyltransferase-like 4; POLG2, Polymerase-gamma2; UNG, Uracil DNA glycosylase; TOP3A, Topoisomerase IIIalpha; OGG1, 8-oxoguanine DNA glycosylase; TFB2M, Mitochondrial transcription factor B2; TOP1MT, Mitochondrial topoisomerase I; RNASEH1, Ribonuclease H1; SSBP1, Single-stranded DNA-binding protein 1; TFAM, Mtochondrial transcription factor A; RECQL4, RecQ protein-like 4; POLRMT, Mitochondrial RNA polymerase; POLG, Polymerase gamma; POLB, Polymerase beta; TWNK, Twinkle mitochondrial DNA helicase; PRIMPOL, Primase-polymerase; MGME1, Mitochondrial genome maintenance exonuclease 1.

follows: mtDNA MRG cluster A (n = 140), mtDNA MRG cluster B (n = 214), and mtDNA MRG cluster C (n = 131) (Fig 2B-D). Prognosis analysis revealed that patients in clusters A and C had better prognoses than did those in cluster B (Fig 2F). In addition, there were significant differences in clinicopathological features (such as TNM stage, sex, age, and survival status) among the three mtDNA MRG clusters (Fig 2E), and most mtDNA MRGs were highly expressed in mtDNA MRG cluster B.

### Biological functions and TME immune cell infiltration features of different mtDNA MRG clusters

GSVA enrichment analysis revealed differences in a variety of pathways among mtDNA MRG clusters A, B and C. mtDNA MRG cluster B was enriched mainly in matrix and carcinogenic activation pathways, such as the mTOR signaling pathway, cell cycle and RNA degradation (Fig 3A), while mtDNA MRG cluster A and mtDNA MRG cluster C were significantly enriched in immune regulation and metabolic pathways, such as the metabolism of fatty acids, arachidonic acid and lysine (Fig 3B-C). Subsequently, the relationships between the mtDNA MRG clusters and immune infiltrating cells were further analyzed, and significant differences in TME cell infiltration were detected among the three mtDNA MRG clusters (Fig 3D). Cluster A had the most infiltrating immune cells and presented enrichment of innate immune cells, including eosinophils, macrophages, and mast cells, while cluster B had the lowest infiltration level. The results of the PCA showed differences between the transcriptome profiles of the three mtDNA MRG clusters (Fig 3E).

### Identification and biological functions of gene clusters

The differences in the mtDNA MRG clusters were analyzed using the limma package. As shown in the figure, there were 8293 DEGs between clusters A-B, 2043 DEGs between clusters A-C, and 9064 DEGs between clusters B-C. The common genes in the intersection of DEGs among the three mtDNA MRG clusters were identified, and 740 common DEGs were considered core genes (Fig 4A). KEGG analysis revealed that the 740 core genes were enriched in carcinogenesis and RNA modification pathways, mainly in the cell cycle, RNA degradation, and DNA base repair (Fig 4B).

Based on 740 mtDNA-related genes, HCC patients were divided into three gene clusters by unsupervised cluster analysis: geneCluster A (n = 120), geneCluster B (n = 153), and geneCluster C (n = 212) (Fig 5A-C). There were significant differences in the expression levels of the three mtDNA MRG gene clusters, HCC patients with different gene clusters displayed different clinicopathological characteristics, and most patients with enrichment of geneCluster B showed better survival status or had clinical stage I-II disease (Fig 5D). Kaplan–Meier survival analysis revealed that HCC patients in geneCluster A had a poor prognosis, while those in geneCluster B had a better prognosis (Fig 5E). Most of the mtDNA MRGs were downregulated in gene cluster B but upregulated in geneCluster A (Fig 5F).

### Construction of the mtDNA score and functional annotation

Due to the individual heterogeneity and complexity of HCC patients, the mtDNA score was established to quantify the mtDNA MRG patterns of individual patients. HCC patients were divided into a high mtDNA score group and a low mtDNA score group, and the "surminer" program was used to obtain the best truncation value (0.05). A Sankey chart was generated to show mtDNA score, survival status, mtDNA MRG clusters and gene cluster correlations (Fig 6A). Immune correlation analysis revealed that the mtDNA score was negatively correlated with the activation of CD8 + T cells, eosinophils

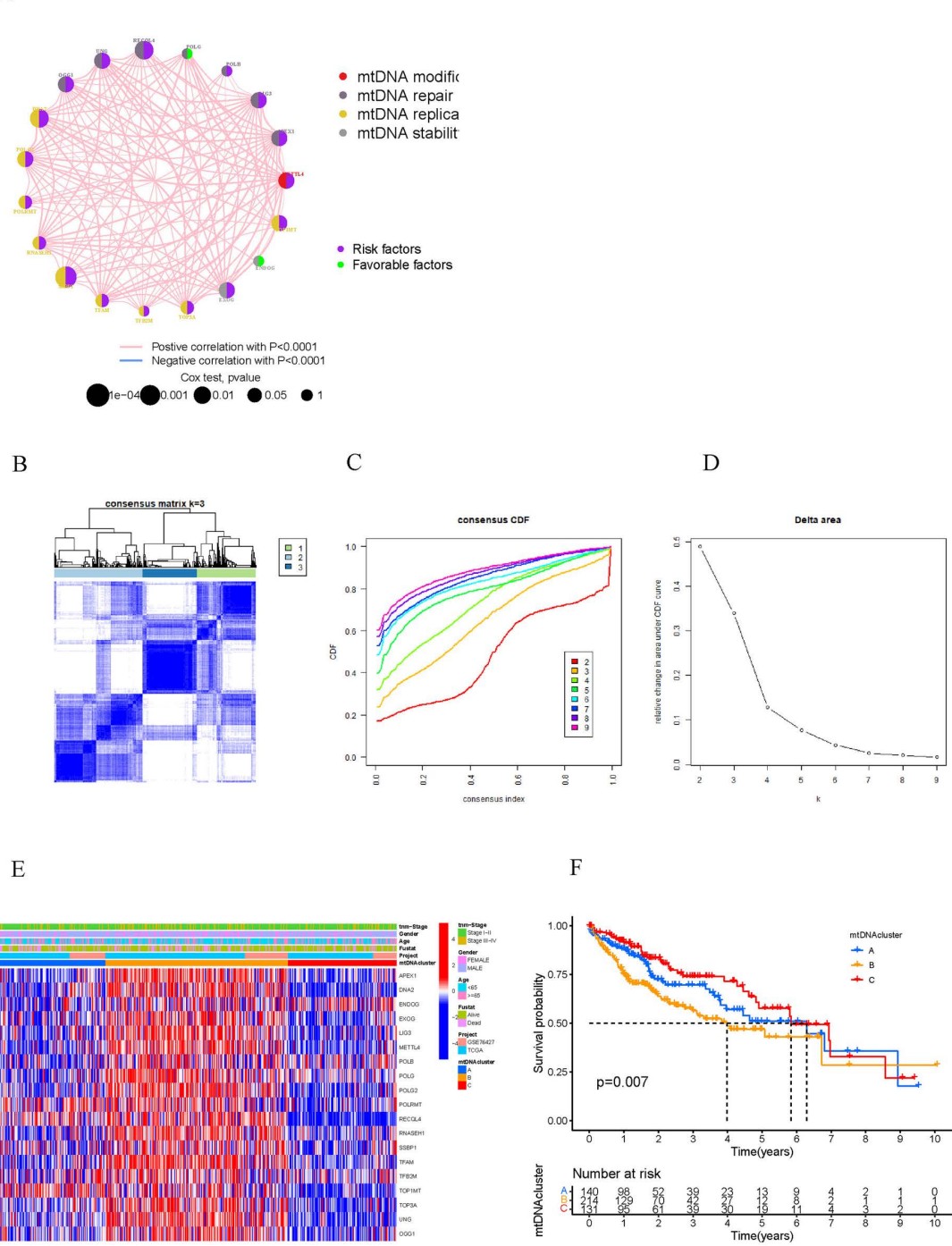

**Fig 2. Consensus clustering of mtDNA MRGs.** A. The size of the circle represents the effect of the mtDNA MRGs on prognosis. The purple and green circles indicate prognostic risk factors and protective factors, respectively. The thickness of the connection lines indicates the strength of the correlation between mtDNA MRGs. Pink represents a positive correlation; blue represents a negative correlation. B. Consensus matrix heatmap of the three clusters (k = 3) and their related regions. C. Area of relative change under the cumulative distribution function curve. D. Relative change region below the cumulative distribution function curve. E. A heatmap annotated with tumor stage, survival status, age, sex and the mtDNA MRG cluster. F. Survival analysis based on mtDNA MRG clusters.

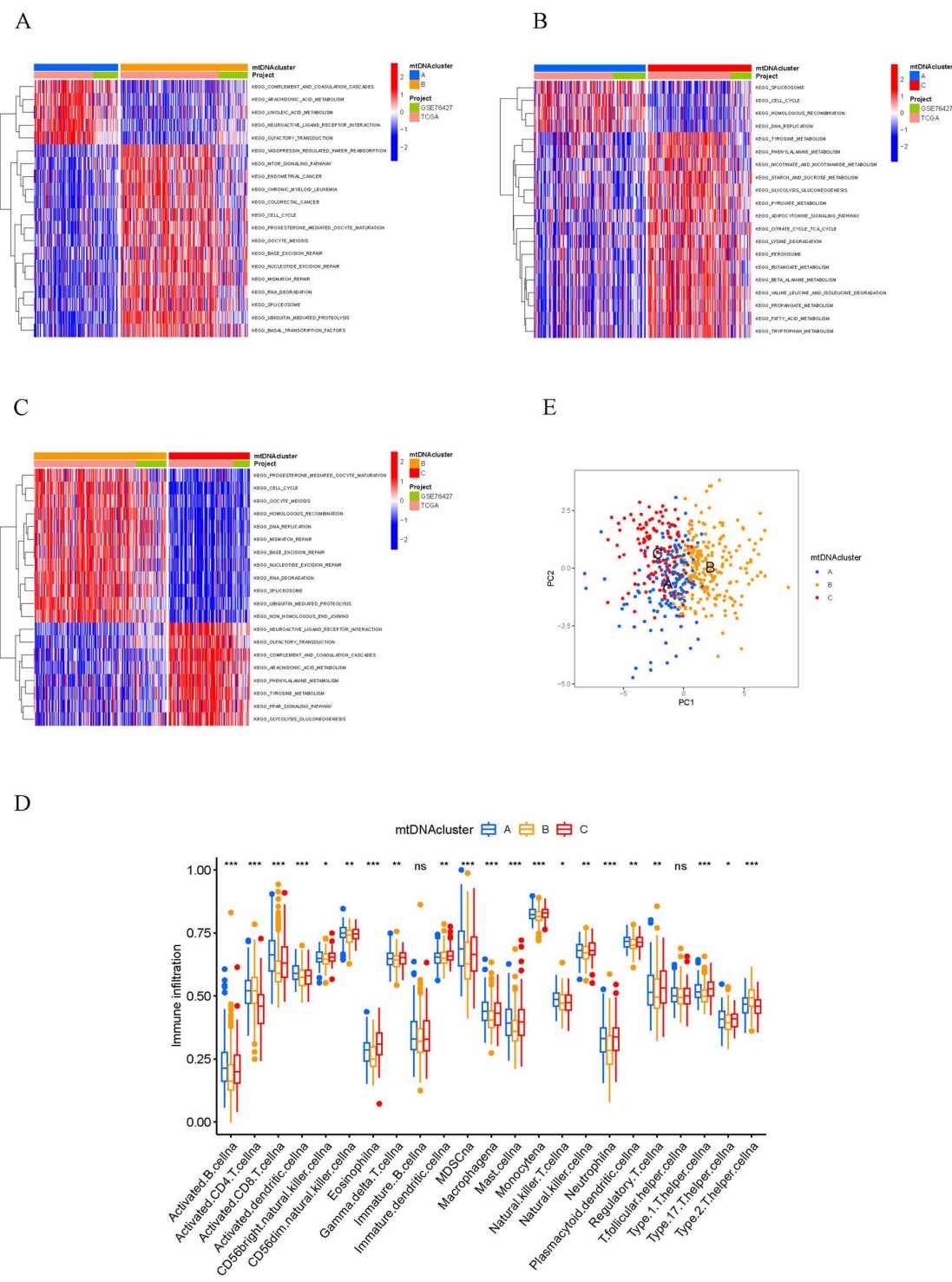

**Fig 3. Biological characteristics and tumor microenvironments of samples enriched in different mtDNA MRG clusters.** Heatmaps showing different biological processes, with red representing activation pathways and blue representing inhibition pathways. A. mtDNA MRG cluster A and mtDNA MRG cluster B. B. mtDNA MRG cluster A and mtDNA MRG cluster C. C. mtDNA MRG cluster B and mtDNA MRG cluster C. D. Abundance of TME-infiltrating cells in samples enriched in the three mtDNA MRG clusters. E. Transcriptome-level differences between mtDNA MRG clusters based on PCA.

A

B

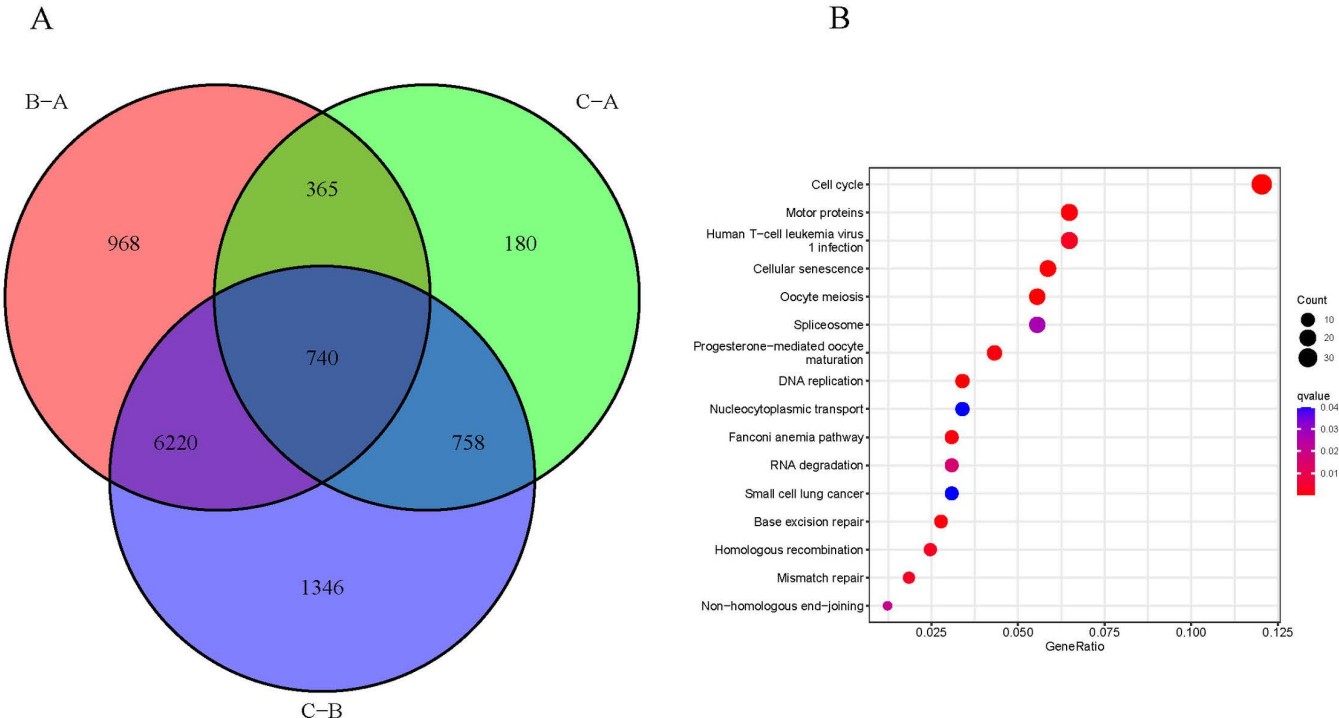

**Fig 4. Identification and functional analysis of DEGs in the mtDNA MRG clusters.** A. DEGs among mtDNA MRG clusters. The red part represents the DEGs between clusters B-A, the green part represents the DEGs between clusters C-A differential gene, and the gray part represents the DEGs between clusters C-B. B. KEGG enrichment analysis of the core genes.

and mast cells but was positively correlated with the activation of CD4 + T cells (Fig 6B). The Kruskal–Wallis test was conducted for the mtDNA MRG cluster, and mtDNA MRG cluster B had the highest score. mtDNA MRG cluster C had the lowest score (Fig 6C); for gene clusters, geneCluster A had the highest score, and geneCluster B had the lowest score (Fig 6D). Survival analysis revealed a significant survival advantage in HCC patients with a low mtDNA score (Fig 6E).

### mtDNA score and TMB

HCC patients were divided into high-TMB and low-TMB groups based on the TMB. Survival analysis revealed that patients with low TMB had better survival than those with high TMB (Fig 7A). When the mtDNA score was combined with the TMB to predict the prognosis of patients with HCC, patients with both high TMB and high mtDNA score had the worst prognosis (Fig 7B). In addition, the waterfall plots show the landscape of tumor somatic mutations between the high- and low-mtDNA-score groups. The most common somatic mutations were in TP53 (58%) in the high mtDNA score group (Fig 7C) and in CTNNB1 (28%) in the low mtDNA score group (Fig 7D).

### Correlation analysis of the mtDNA score with clinicopathological features and immunotherapy efficacy

According to the clinicopathological feature correlation analysis, younger age (< 65 years), stage III-IV, and death at the clinical endpoint were significantly associated with a higher mtDNA score (Fig 8A-C). Stratified analysis revealed that patients in the low mtDNA score group had a better prognosis than patients in the high mtDNA score group (Fig 8D-F). These results further supported that the mtDNA score might be used to predict the prognosis of HCC.

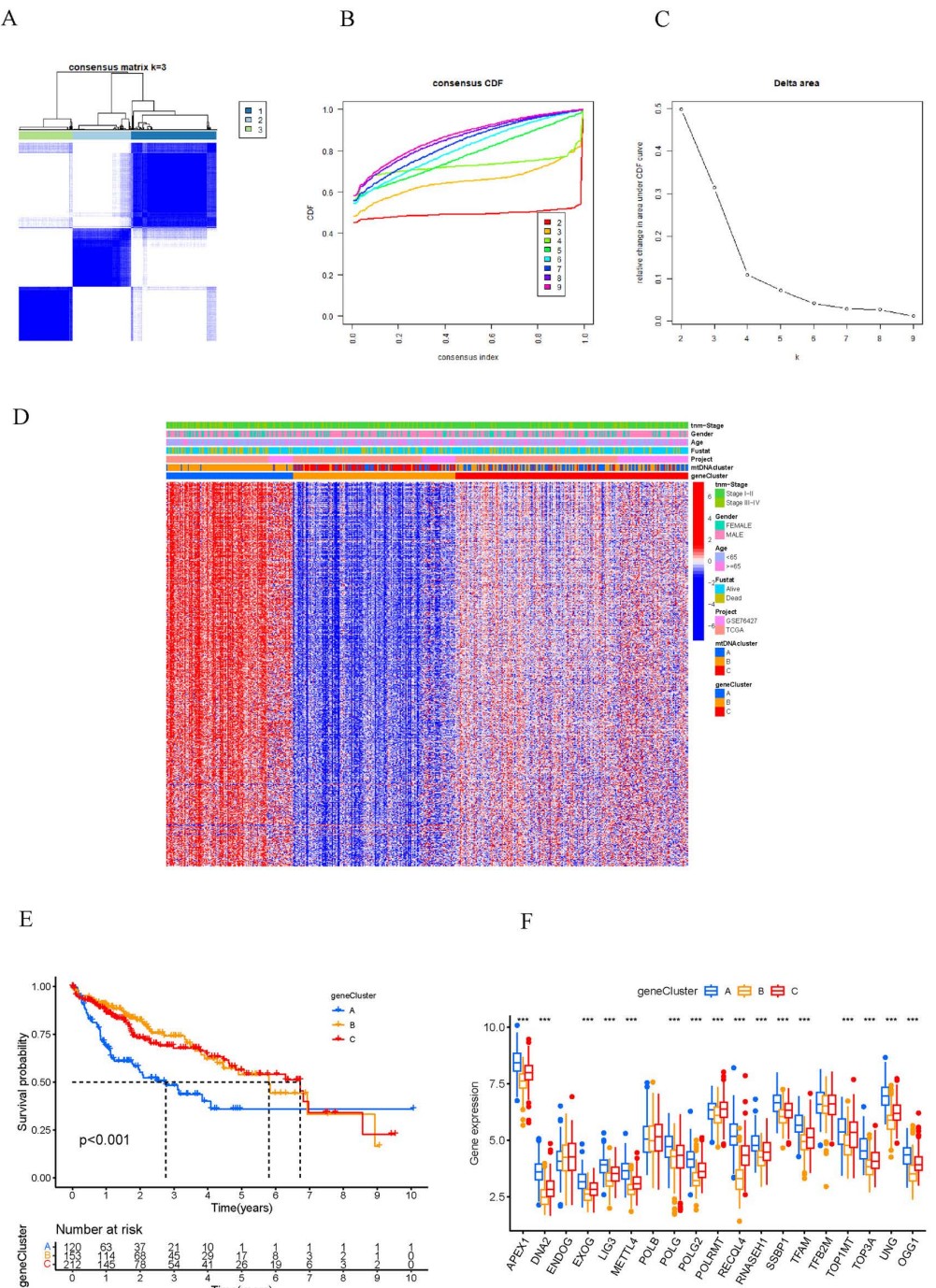

**Fig 5. Three mtDNA MRG gene clusters in HCC patients.** A-C. Consensus clustering of genecluster for k = 3. D. Survival analysis of three gene clusters. E. Heatmaps showing the gene profiles of the three gene clusters and the relationships between the gene clusters and clinicopathological features. F. Expression levels of mtDNA MRGs in three gene clusters. APEX1, Apurinic/apyrimidinic endonuclease 1; DNA2, ATP-dependent helicase/nuclease 2; EXOG, Endo/exonuclease G-like; EXDOG, endonuclease G; LIG3, ligase III; METTL4, Methyltransferase-like 4; POLG2, Polymerase-gamma2; UNG, Uracil DNA glycosylase; TOP3A, Topoisomerase IIIalpha; OGG1, 8-oxoguanine DNA glycosylase; TFB2M, Mitochondrial transcription factor B2; TOP1MT, Mitochondrial topoisomerase I; RNASEH1, Ribonuclease H1; SSBP1, Single-stranded DNA-binding protein 1; TFAM, Mtochondrial transcription factor A; RECQL4, RecQ protein-like 4; POLRMT, Mitochondrial RNA polymerase; POLG, Polymerase gamma; POLB, Polymerase beta.

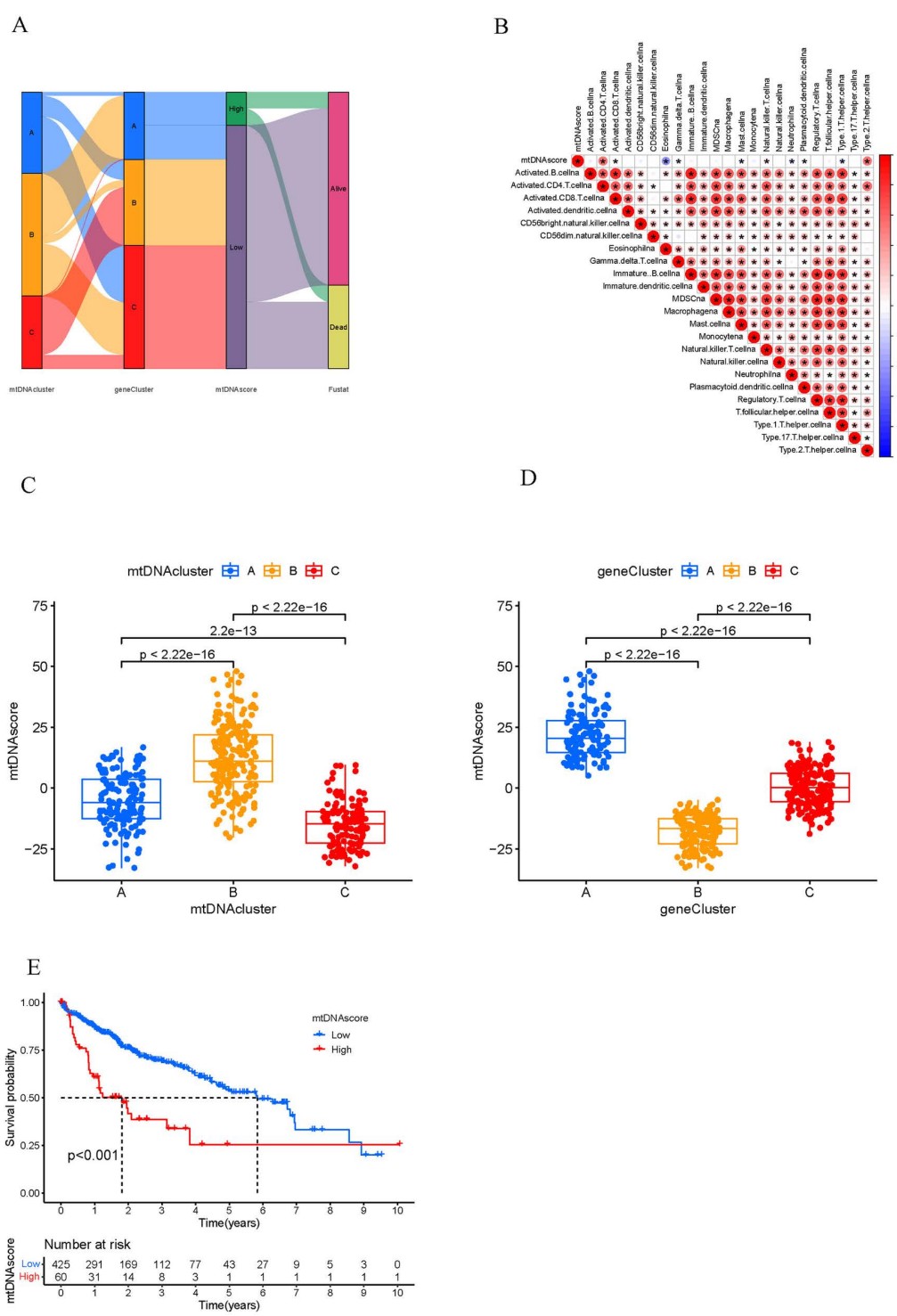

**Fig 6. Construction and functional annotations of mtDNA score signatures.** A. Alluvial maps showing changes in the mtDNA MRG cluster, gene cluster, mtDNA score and survival status. B. Correlation between the DNA score and immune cell infiltration. C. mtDNA score differences among the three mtDNA MRG clusters. D. mtDNA score differences among the three mtDNA MRG gene clusters. E. Survival analysis of patients with high mtDNA scores and low mtDNA scores.

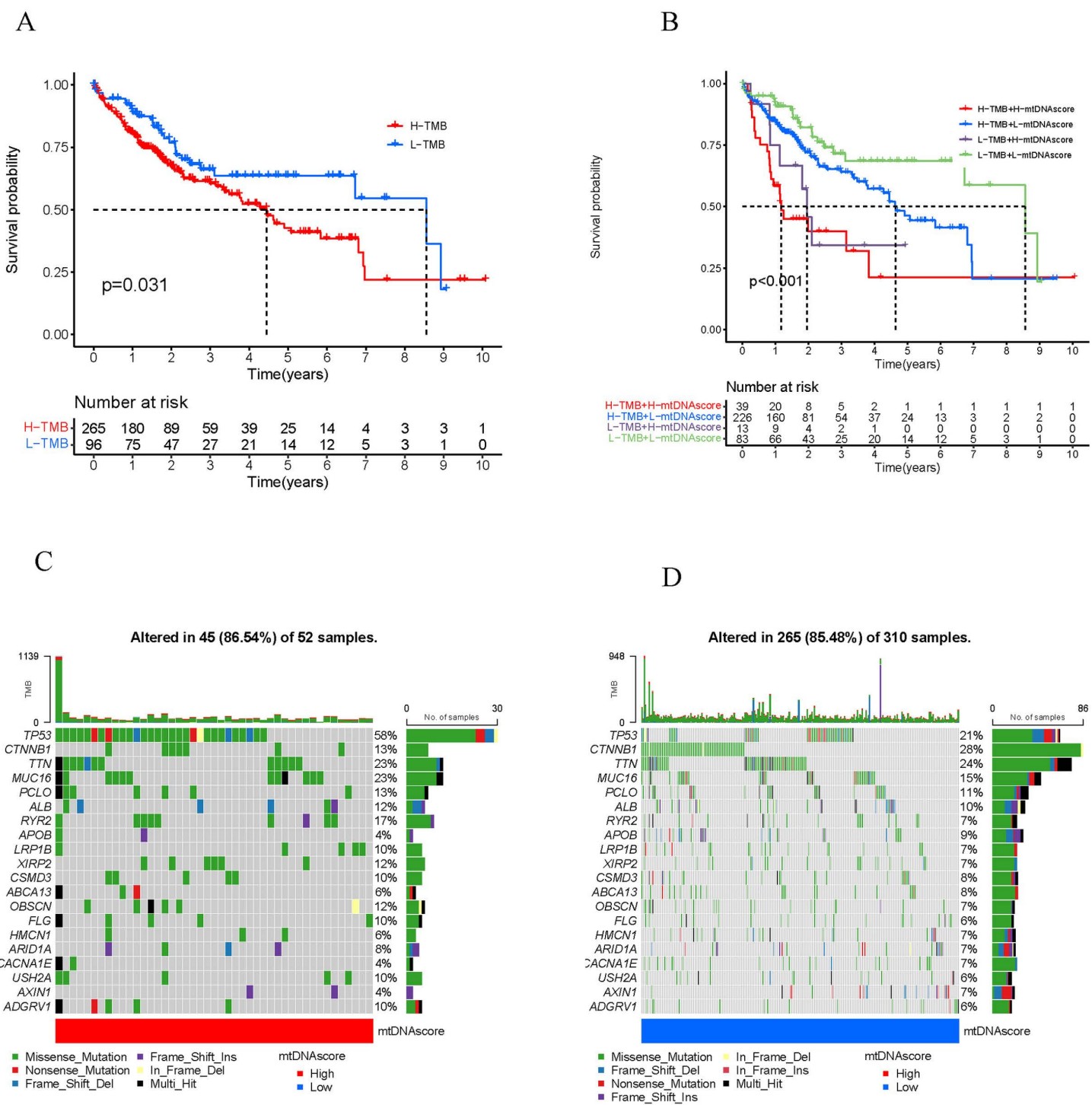

**Fig 7. mtDNA score and tumor somatic mutations.** A. Survival analysis of patients with high and low TMB. B. Survival analysis of patients stratified by TMB combined with the mtDNA MRG score. C. Waterfall plot for the high mtDNA MRG score group. D. Waterfall map for the low mtDNA MRG score group. TMB, tumor mutation burden.

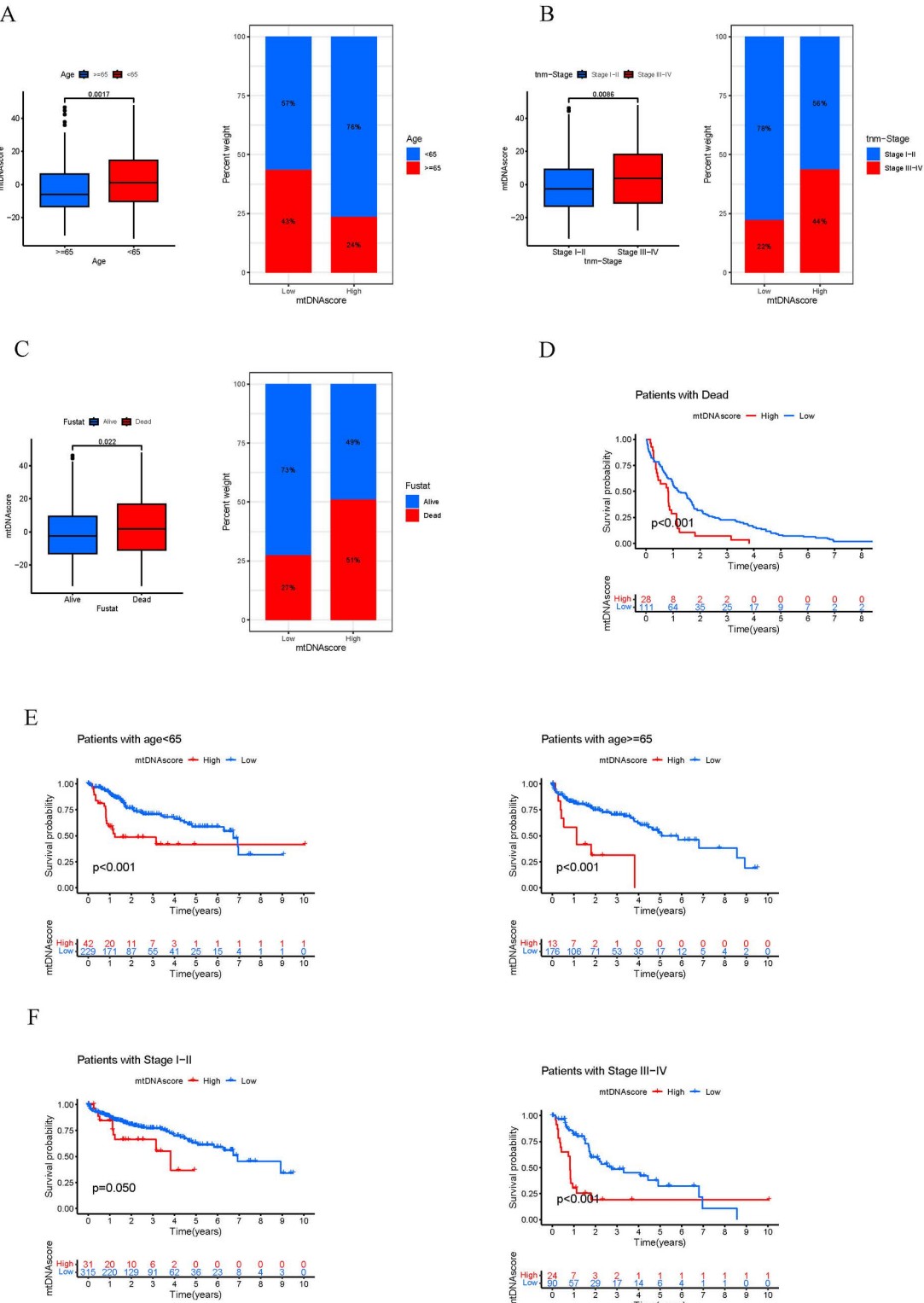

**Fig 8. Analysis of correlation between the mtDNA MRG score and prognosis in different clinical subgroups.** A-C. Correlation analysis between clinical features and the mtDNA MRG score. D-F. Associations of the mtDNA MRG score with death outcome (D), age (E), and stage **(F)**.

To further explore the correlation between the mtDNA score and immunotherapy efficacy, the therapeutic response to ICIs, such as CTLA-4/PD-1 inhibitors, was analyzed. Fig 9A shows no difference in PD-L1 expression between the high mtDNA score group and the low mtDNA score group (Fig 9A). The anti-CTLA-4 and anti-PD-1 response rates were different between the high mtDNA score group and the low mtDNA score group (p = 0.018) (Fig 9B). However, there was no difference between the high mtDNA score group and the low mtDNA score group in terms of the response to anti-PD-1, anti-CTLA-4, or combined anti-CTLA-4/PD-1 immunotherapy (Fig 9C-E).

## Discussion

In this study, 22 mtDNA MRGs and three mtDNA MRG clusters were identified. HCC patients with enrichment of the three mtDNA MRG clusters had different prognoses and TME immune cell infiltration characteristics. mtDNA MRG cluster A is characterized by enhanced tumor matrix activity and abundant innate immune cell infiltration, corresponding to the immune rejection phenotype; mtDNA MRG cluster B is characterized by immunosuppression, corresponding to the immune-desert phenotype; and mtDNA MRG cluster C is characterized by adaptive immune cell infiltration and immune activation, corresponding to the immunoinflammatory phenotype [29]. The immune rejection and immune-desert phenotypes are considered to indicate noninflammatory tumors, primarily tumors that lack immune cell invasion in the parenchyma and stroma, rarely express PD-L1, are located at the opposite end of the tumor immune continuum, and histologically lack immune invasion and antigen presentation (low MHC class I), while exhibiting high tumor cell proliferation. The immunoinflammatory phenotype indicates inflammatory tumors, in which the TME has a high degree of infiltration of immune cells, such as T cells, CD8 + T cells producing IFN-γ, and PD-1-positive immune cells [30]. There were significant differences in the TME features among samples enriched in the three mtDNA MRG clusters. The GSVA results showed that mtDNA MRG clusters A and C were mainly enriched in pathways related to tumor metabolism, leading to relative inhibition of tumor growth and good prognosis [31], while mtDNA MRG cluster B was mainly enriched in the cell cycle, RNA degradation, DNA replication, base repair and other pathways, which are closely related to cancer progression [32–34]. Therefore, HCC patients with enrichment of mtDNA MRG cluster B have a poor survival prognosis.

Analysis of mtDNA MRG clusters is helpful for understanding the TME infiltration characteristics. Due to individual heterogeneity and specificity, a scoring model was constructed to evaluate the mtDNA MRG patterns in individual HCC patients. The mtDNA score was higher for the immune-desert phenotype and lower for the immune inflammatory phenotype. A high mtDNA score is associated with increased immune and stromal cell infiltration [14]. This result suggested that the mtDNA score is a useful tool for comprehensively evaluating the enrichment of mtDNA MRGs in individual tumors.

Mitochondria are important organelles, and mtDNA variants (with mutations/single nucleotide polymorphisms) and disorders of mitochondrial coding genes are associated with cancer progression. The genes with the highest mutation frequencies in HCC patients with high and low mtDNA scores were TP53 and CTNNB1, respectively. TP53 encodes the P53 protein known as a canonical tumor suppressor, and protein isoforms derived from mutated p53 are usually overexpressed in cancer; these isoforms lose their tumor suppressor function and promote tumorigenesis [35]. p53 promotes mitochondrial respiration in a variety of ways. Mutation of p53 can affect the morphology and structure of mitochondria, resulting in loss of mitochondrial function. Mutated p53 affects mitochondrial energy production by regulating mtDNA replication and maintenance, assembly of the mitochondrial respiratory chain complex and regulation of the mitochondrial membrane potential [36]. HCC is a highly heterogeneous tumor, and mutations in the TP53 gene are involved in the development of intratumor heterogeneity and may contribute to treatment failure and drug resistance in many HCC patients [37,38]. In the present study, the poor prognosis of HCC in the high mtDNA score group was associated with excessive TP53 mutation [39]. The probability of CTNNB1 mutation was greater in HCC patients with low mtDNA scores, and CTNNB1 mutation in HCC patients was associated with specific well-differentiated HCC subtypes, in contrast to the findings in TP53-mutated HCC patients, in whom histologically poorly differentiated, highly proliferative, large trabecular clumps developed [40]. HCC patients with high mtDNA scores had a significantly greater probability of having genetic

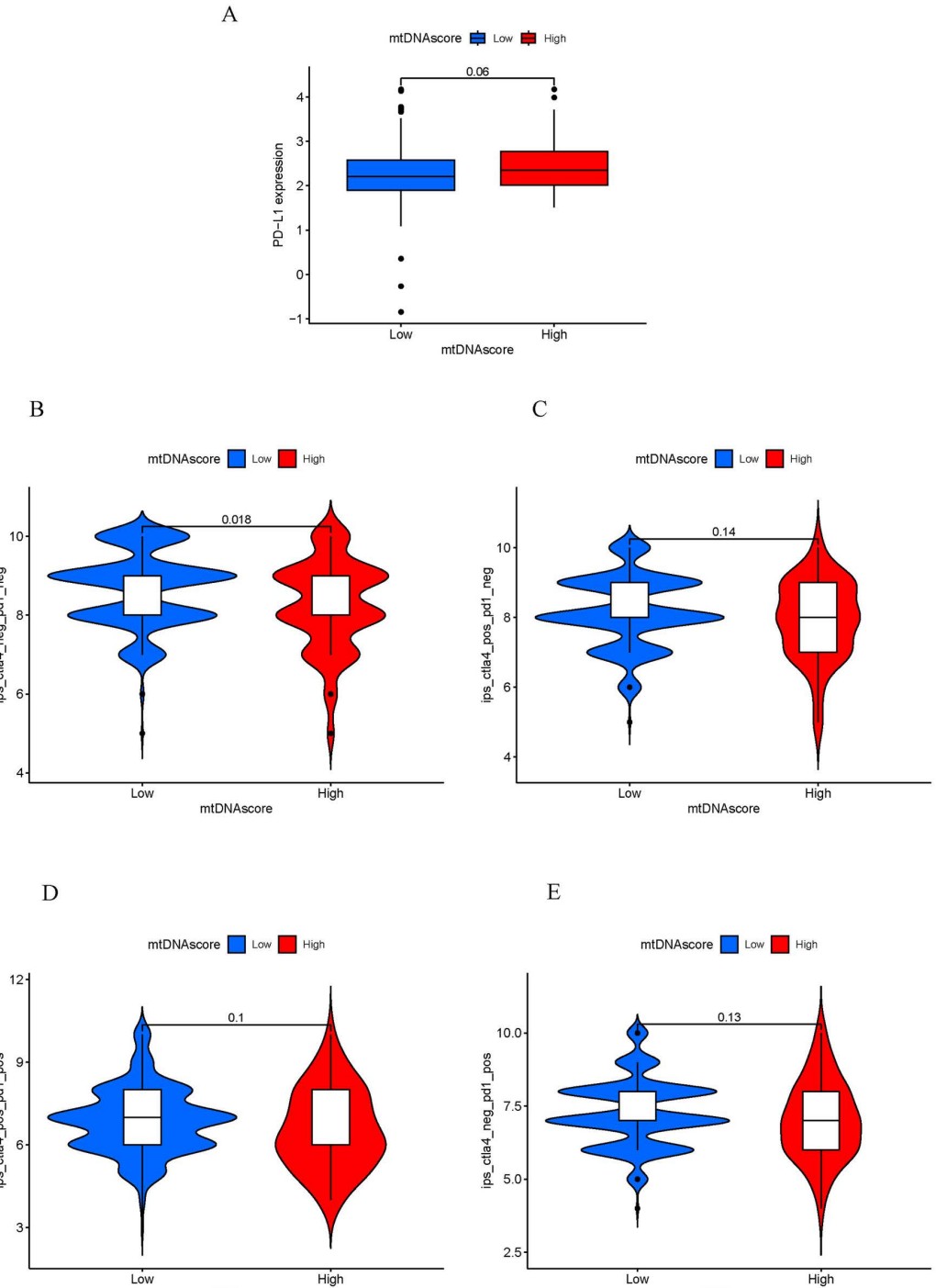

**Fig 9. Correlations between the mtDNA score and immunotherapy response.** A. Expression of PD-L1 in the low-mtDNA score group and high-mtDNA score group. B. Analysis of differences in the response to CTLA-4 negative and PD-1 negative therapy between the low mtDNA score group and high mtDNA score group. C. Analysis of the difference in anti-PD-1 immunotherapy efficacy between the low mtDNA score group and the high mtDNA score group. D. Analysis of the difference in anti-CTLA-4 immunotherapy efficacy between the low mtDNA score group and the high mtDNA score group. E. Differences in the efficacy of anti-PD-1/anti-CTLA-4 combined immunotherapy between the low-mtDNA MRG score group and the high-mtDNA MRG score group. PD-1, Programmed cell death-1; PD-L1, Programmed death ligand 1; CTLA-4, Cytotoxic T-lymphocyte antigen 4.

mutations than did those with low mtDNA scores. TMB, the total number of mutations in each coding region of the tumor genome, is a new indicator for predicting immune responses in various cancers [41]. The finding that patients with high TMB and mtDNA scores had the worst prognosis may be related to genomic instability caused by a high frequency of TP53 mutations and poor HCC differentiation [42].

Previous studies have shown that mitochondrial dysfunction is a key factor in cancer pathogenesis [43]. Dysregulation of mitochondria plays an important role in oncogenesis and progression of HCC [44,45]. However, there are few reports on the relationship between mitochondrial genes and the development of HCC. In our data, HCC patients with high expression of RECQL4 and APEX1 have a worse prognosis. A retrospective study has shown that the expression of RECQL4 mRNA in HCC tissues is significantly higher than that in adjacent normal liver tissues. Overexpression of RECQL4 may give hepatocellular carcinoma cells unlimited proliferation potential and promote hepatocellular tumor occurrence [46]. In addition, APEX1 was significantly upregulated and predicted poor clinical overall survival in HCC patients. Silencing APEX1 inhibited the proliferation of HCC cells in vivo and in vitro, and it repressed invasion and migration [47]. So targeted mitochondrial gene therapy is an attractive strategy to inhibit HCC progression.

Analysis of the correlation between the mtDNA score and clinical data revealed that patients aged < 65 years who died at the clinical endpoint and with stage III-IV disease had relatively high mtDNA scores and poor clinical outcomes. Many studies have shown that the copy number and function of mtDNA decrease with age [48]. An increase in mtDNA copy number leads to mitochondrial dysfunction, thus causing disease. Patients with malignant tumors such as lung cancer, pancreatic cancer and kidney cancer have increased mtDNA copy numbers [49]. Therefore, patients aged < 65 years have higher mtDNA scores than patients aged > 65 years.

The mtDNA score can also be used to evaluate certain clinical characteristics of patients. The effects of anti-CTLA4 and anti-PD-1 treatments differed between the high mtDNA score group and the low mtDNA score group. The value of the mtDNA score in predicting the outcome of PD-L1/CTLA-4 immunotherapy was subsequently further analyzed; however, the mtDNA score did not significantly differ between responders and nonresponders receiving PD-L1, CTLA-4, or combination therapy. These results suggest that an imbalance in mtDNA MRGs affects immunotherapy resistance in HCC patients. Drug resistance is a major obstacle to cancer treatment [50]. Mitochondrial metabolic plasticity contributes to resistance in most types of anticancer therapy [51]. In melanoma, alterations in mitochondrial function may affect the energy metabolism, apoptosis, and immunotherapy response of melanoma cells, thus affecting the progression of melanoma and resistance to PD-1 inhibitors [52]. Therefore, the results of this study can guide future studies on the mechanisms related to immunotherapy resistance and mtDNA MRGs in HCC.

Although we identified 22 mtDNA MRGs, using retrospective data to construct mtDNA scores. In clinical practice, mtDNA scores may be useful in predicting clinical outcomes as well as evaluating the corresponding TME cell infiltration characteristics and TMB in individual HCC patients. However, several limitations should be acknowledged. First, to ensure data robustness, we excluded samples located far from the main cluster, as these may represent biological outliers deviating from the predominant group trend. Given that all specimens were derived from human tissues, inter-individual variability is expected; nevertheless, the overall differences remained within an acceptable range. Moreover, the TCGA and GEO datasets employed in this study have been extensively validated in prior research, supporting their reliability. To minimize technical artifacts, we performed rigorous data correction, ensuring that subsequent findings reflect true biological characteristics rather than batch effects. Second, all clinical data in this study came from public databases, and the mtDNA MRG score established on this basis may have potential biases. In addition, further cell and animal experiments are needed to explore the functional role of mtDNA MRGs in HCC, which could help provide stronger clues to guide clinical applications.

## Conclusion

In conclusion, a comprehensive and systematic analysis of mtDNA MRGs was conducted in this study. The mtDNA score established based on mtDNA MRGs can enhance the understanding of the cell infiltration characteristics of the TME and

provide innovative ideas regarding potential therapeutic targets and more effective immunotherapy strategies to provide theoretical guidance for treatment decision making and prognosis prediction in HCC patients.

## Supporting information

**S1 File. The performance of oPLS-DA model was assessed by R²Y (goodness of fit) and Q² (goodness of prediction) through cross-validation, along with permutation testing to evaluate robustness.**
(DOCX)

## Author contributions

**Data curation:** Fan Rong.

**Software:** Bin Cheng.

**Supervision:** Zhongji Meng.

**Visualization:** Ling Guo, Shaobo Zeng, Xunliang Xu, Zhongji Meng.

**Writing – original draft:** Fan Rong.

**Writing – review & editing:** Bin Cheng, Ling Guo, Shaobo Zeng, Xunliang Xu, Zhongji Meng.

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
