## [Decision Letter · Decision Letter 0]

5 Aug 2024

PONE-D-24-22922Correlation analysis of mitochondrial DNA maintenance-related genes with HCC prognosis, tumor mutation burden and tumor microenvironment featuresPLOS ONE

Dear Dr. Rong,

Thank you for submitting your manuscript to PLOS ONE. After careful consideration, we feel that it has merit but does not fully meet PLOS ONE’s publication criteria as it currently stands. Therefore, we invite you to submit a revised version of the manuscript that addresses the points raised during the review process.

**Review of PONE-D-24-22922**

**Please define the mtDNA score, TMB and TME (lines 25-17).****I suggest you use the keyword mitochondrion and not mitochondrial (line 44).****“own superhelical double-linked 61 ring of genetic material” is unnecessary (line 60).****Move gene list from lines 115-118 to a separate table.****Describe ssGSEA in more detail in the Methods section.****What is this data set from Charoenteng? Add a reference to it.****Why was an adjusted p-value of 0.001 used? (line 143)****Lines 166-176: indentation is problematic****You mention 8293 DEGs between clusters A and B and 2043 DEGs between clusters A and C, and 9064 between clusters B and C. Why are there so many? What was the fold change cutoff limit? What would happen if you increase this cutoff limit?****Starting from line 317 you describe the role of TP53 in cancer. This is well known. What is the novelty of your findings? As you write, HCC is very heterogeneous, what other genes play a role in HCC that you found?**

We look forward to receiving your revised manuscript.

Kind regards,

Matthew Cserhati, Ph.D

Academic Editor

PLOS ONE

Journal Requirements:

3. Thank you for uploading your study's underlying data set. Unfortunately, the repository you have noted in your Data Availability statement does not qualify as an acceptable data repository according to PLOS's standards.

**Additional Editor Comments:**

Review of PONE-D-24-22922

1. Please define the mtDNA score, TMB and TME (lines 25-17).

2. I suggest you use the keyword mitochondrion and not mitochondrial (line 44).

3. “own superhelical double-linked 61 ring of genetic material” is unnecessary (line 60).

4. Move gene list from lines 115-118 to a separate table.

5. Describe ssGSEA in more detail in the Methods section.

6. What is this data set from Charoenteng? Add a reference to it.

7. Why was an adjusted p-value of 0.001 used? (line 143)

8. Lines 166-176: indentation is problematic

9. You mention 8293 DEGs between clusters A and B and 2043 DEGs between clusters A and C, and 9064 between clusters B and C. Why are there so many? What was the fold change cutoff limit? What would happen if you increase this cutoff limit?

10. Starting from line 317 you describe the role of TP53 in cancer. This is well known. What is the novelty of your findings? As you write, HCC is very heterogeneous, what other genes play a role in HCC that you found?

Reviewers' comments:

Reviewer's Responses to Questions

**Comments to the Author**

1. Is the manuscript technically sound, and do the data support the conclusions?

Reviewer #1: Yes

Reviewer #2: Partly

2. Has the statistical analysis been performed appropriately and rigorously? 

Reviewer #1: Yes

Reviewer #2: I Don't Know

3. Have the authors made all data underlying the findings in their manuscript fully available?

Reviewer #1: Yes

Reviewer #2: No

4. Is the manuscript presented in an intelligible fashion and written in standard English?

Reviewer #1: Yes

Reviewer #2: No

5. Review Comments to the Author

**Reviewer #1:**  The manuscript titled "Correlation analysis of mitochondrial DNA maintenance-related genes with HCC prognosis, tumor mutation burden, and tumor microenvironment features" presents a comprehensive analysis of the relationship between mitochondrial DNA (mtDNA) maintenance-related genes (mtDNA MRGs) and the prognosis of hepatocellular carcinoma (HCC), tumor mutation burden (TMB), and tumor microenvironment (TME) features. The study utilized a dataset of 487 HCC samples from The Cancer Genome Atlas (TCGA) and Gene Expression Omnibus (GEO) to identify 22 mtDNA MRGs through screening and then applied unsupervised clustering and principal component analysis (PCA) to construct an mtDNA score model. This model was used to examine associations with clinicopathological characteristics, TMB, TME cell infiltration, and biological processes.

Methodological Considerations: The study design appears sound, utilizing a large sample size and robust bioinformatic tools. However, there are several areas that could be improved upon for greater clarity and rigor:

1. Sample Selection and Control Group: While the use of TCGA and GEO datasets is commendable, the inclusion criteria for selecting HCC samples should be more clearly described. Moreover, the control group's characteristics should also be provided for comparison purposes.

2. Screening of mtDNA MRGs: The process of identifying the 22 mtDNA MRGs requires further detail. How were these genes prioritized over others? What criteria were used for screening?

3. Unsupervised Clustering: Details about the machine learning model employed for unsupervised clustering need to be elaborated upon, including parameters and validation methods.

4. PCA Model Construction: The construction of the mtDNA score model using PCA should be detailed, including how the principal components were selected and validated.

Statistical Analysis: The statistical methods used appear appropriate for the data type and research question. Nevertheless, the following points should be addressed:

1. Statistical Power: It would be beneficial to include a discussion on the statistical power of the study given the sample size and the number of variables being analyzed.

2. Multiple Testing Correction: With multiple comparisons being performed, a correction method such as Bonferroni or False Discovery Rate (FDR) should be applied and reported.

3. Association Strength and Significance: The strength and significance of correlations should be reported comprehensively, including effect sizes and confidence intervals.

Results Interpretation: The findings of significant differences in mtDNA MRG expression between HCC samples and normal controls, as well as the identification of three molecular subtypes with distinct clinical and immune profiles, are intriguing. However, the interpretation of results should be cautious and conservative:

1. Biological Relevance: The biological implications of the molecular subtypes and their correlation with TME features need to be discussed in the context of existing literature.

2. Causal Inference: The observational nature of the study limits the ability to infer causality. Any discussion of causative mechanisms should be framed within this limitation.

3. External Validation: Although not feasible in this study, future directions should include external validation of the mtDNA score model in independent cohorts.

Discussion: The discussion should address the limitations of the study and consider the broader implications of the findings. It is important to discuss how the results fit into the current understanding of mtDNA MRGs in HCC and what further research is needed.

In conclusion, while the manuscript provides valuable insights into the role of mtDNA MRGs in HCC, several methodological and analytical improvements are necessary for publication. The authors should address these concerns thoroughly in a major revision before the manuscript can be considered for acceptance.

**Reviewer #2: ** Title: Correlation analysis of mitochondrial DNA maintenance-related genes with HCC prognosis, tumor mutation burden and tumor microenvironment features

This paper presents a study to focus on exploring the important role of Mitochondrial DNA maintenance-related genes (mtDNA MRGs) in the hepatocellular carcinoma (HCC). Authors proposed mtDNA score model which can be used to assess tumor immune cell invasion, clinicopathological features, genetic variation, and immunotherapy response. However, there are questions that limit my enthusiasm for the paper, as outlined below.

- Data section needs to be improved

o Add more details regarding the batch correction method used.

o Include a figure showing data before and after batch correction (as supplementary material).

o Specify the type of RNA data used (RNA-seq or microarray).

o Clarify the type of normalization applied to the data.

- The Method section needs to be improved. Authors mentioned the packages used without providing details about the methods. It should not be assumed that all readers are familiar with or can recall the methods implemented in each package (e.g., limma, ConsensusClusterPlus).

- The clustering step using samples and genes is not clear and easy to follow. Please add details of the method, including the type of clustering (e.g., hierarchical clustering) and the distance metric used.

- Additionally, add dendrograms to the heatmaps (e.g., Figures 2E and 5D) to better illustrate the clusters found and align with the findings.

- The mtDNA score model is not clear. Please specify on which subset of genes the PCA analysis was applied. Additionally, Figure 3 related to score modeling and TME was not shared and is missing.

- What does the y-axis label show in Figures 9B-E? Is it the IPS signature score? Additionally, there are more ICB signatures or biomarkers to consider, including PDCD1, CD274, PDCD1LG2, CTLA4, TIM3, LAG3, TIGIT, and resources like bhklab/SignatureSets: Compendium of published molecular signatures (github.com) and/or https://pubmed.ncbi.nlm.nih.gov/36055464/

- A GitHub repository or another container to access all the code is needed.

- Validation is highly recommended.

6. PLOS authors have the option to publish the peer review history of their article (what does this mean? ). If published, this will include your full peer review and any attached files.

**Do you want your identity to be public for this peer review?** For information about this choice, including consent withdrawal, please see our Privacy Policy .

Reviewer #1: No

Reviewer #2: No

---

## [Author Response · Author response to Decision Letter 1]

12 Oct 2024

Response to Editor

Thanks for your kind suggestions. We have read your suggestions carefully and try our best to give the following response:

Q1. Please define the mtDNA score, TMB and TME (lines 25-17).

Response: Thank you for your suggestions. The tumor microenvironment (TME) was defined (lines 73-75 in the revised manuscript). mtDNA score was defined (lines 185-186 in the revised manuscript). Tumor mutational burden (TMB) reflects cancer mutation quantity (doi:10.1016/j.ccell.2020.10.001), now TMB is emerging as a potential biomarker (doi:10.1093/annonc/mdy495). As a common name, in many articles, TMB is directly cited (doi: 10.1016/j.ccell.2023.09.006; doi:10.7150/thno.52717).

Q2. I suggest you use the keyword mitochondrion and not mitochondrial (line 44).

Response: Thank you for your suggestion. the key word “mitochondrial” has been replaced with “mitochondrion”.

Q3.“own superhelical double-linked 61 ring of genetic material” is unnecessary (line 60)

Response: Thank you for your suggestion. The sentence “A unique feature of mitochondria is that they have their own superhelical double-linked ring of genetic material, called mitochondrial DNA (mtDNA)” has been deleted.

Q4. Move gene list from lines 115-118 to a separate table.

Response: Thank you for your suggestion. If we move gene list to a separate table, there's not much in this table. So, we categorized these genes according to their functions. Maybe it's more categorical and concise. mtDNA replication (DNA2, EXOG, LIG3, POLG, POLG2, POLRMT, RNASEH1, SSBP1, TFAM, TFB2M, TOP1MT, TOP3A, TWNK)�mtDNA repair (APEX1, EXOG, LIG3, OGG1, POLG, PRIMPOL, RECQL4, UNG), mtDNA modifications (METTL4), mtDNA stability and decay (ENDOG, EXOG, MGME1) (lines 125-128 in the revised manuscript).

Q5. Describe ssGSEA in more detail in the Methods section.

Response: Thank you for your comments. The following sentence has been added the Methods section to describe ssGSEA in more detail. ssGSEA was used to calculate the standardized enrichment score via the GSVA package50.The relative abundance of each immune cell type was represented by an enrichment score in ssGSEA analysis and normalized to unity distribution from 0 to 1(doi: 10.7150/thno.52717)(lines 167 to 169).

Q6. What is this data set from Charoenteng? Add a reference to it.

Response: Thank you for your suggestion. Pornpimol Charoentong, et al analyzed tumor-immune cell interactions in 20 solid cancers, revealing the relationship between genotype and immunophenotype and created The Cancer Immunome Atlas (https://tcia.at/). Using machine learning, the determinants of tumor immunogenicity were identified and a quantitative scoring scheme called immunophenotypic scoring was developed. The according reference (doi: 10.1016/j.celrep.2016.12.019) has been added (line 156).

Q7. Why was an adjusted p-value of 0.001 used? (line 143)

Response: We referred to many similar literatures and found that using an adjusted p-value of 0.001 is more reasonable (doi: 10.1186/s12943-020-01170-0; doi: 10.3389/fcell.2022.842220).

Q8. Lines 166-176: indentation is problematic

Response: Thank you for your comment. All the indentation has been deleted in the revised manuscript.

Q9. you mention 8293 DEGs between clusters A and B and 2043 DEGs between clusters A and C, and 9064 between clusters B and C. Why are there so many? What was the fold change cutoff limit? What would happen if you increase this cutoff limit?

Response: Although the consensus clustering algorithm based on mtDNA MRGs expression classified HCC patients into three mtDNA MRG phenotypes, the underlying genetic alterations and expression perturbations within these phenotypes were not well known, so there are so many related genes. P-values were adjusted according to the false discovery rate (FDR), and the P-value less than 0.001 was taken as the screening criteria. If we increase this cutoff limit, more DEGs related genes will be excluded.

Q10. Starting from line 317 you describe the role of TP53 in cancer. This is well known. What is the novelty of your findings? As you write, HCC is very heterogeneous, what other genes play a role in HCC that you found?

Response: The novelty of our findings lies in the following aspects. 1) In the high mtDNA score group, TP53 mutation frequency is the highest, and the poor prognosis of HCC patients in the High mtDNA score group may be related to the high TP53 mutation. 2) Among HCC patients with lower mtDNA scores, HCC patients with TP53 mutations had worse clinicopathological features compared to CTNNB1-mutated HCC patients.

Besides TP53, two genes RECQL4 and APEX1 have been found play a role in HCC, HCC patients with higher expression of RECQL4 and APEX1 have a worse prognosis according with the previous studies (doi:10.3892/ol.2018.78600; doi:10.1016/j.immuni.2019.12.011). In our data, RECQL4 and APEX1 are highly expressed in HCC samples, and the higher expression in RECQL4 and APEX1, the worse prognosis of the HCC patients, which is consistent with the results of other studies. So, our research is feasible.

Response to Reviewer 1

Reviewer 1

Methodological Considerations: The study design appears sound, utilizing a large sample size and robust bioinformatic tools. However, there are several areas that could be improved upon for greater clarity and rigor:

Q1. Sample Selection and Control Group: While the use of TCGA and GEO datasets is commendable, the inclusion criteria for selecting HCC samples should be more clearly described. Moreover, the control group's characteristics should also be provided for comparison purposes.

Response: Dear reviewer, thanks for your kind suggestions. HCC samples in TCGA were collected from patients pathologically diagnosed with hepatocellular carcinoma (HCC) after surgical resection and who had not previously received treatment (ablation, chemotherapy, or radiotherapy) specific to their disease. Cases were staged according to the American Joint Committee on Cancer (AJCC). Each frozen primary tumor specimen had a companion normal tissue specimen (blood or blood components, including DNA extracted at the tissue source site). Adjacent tissue was submitted for some cases. Hematoxylin and eosin (H&E) stained sections from each sample were subjected to independent pathology review to confirm that the tumor specimen was histologically consistent with the allowable hepatocellular carcinomas and the adjacent tissue specimen contained no tumor cells. Adjacent tissue with cirrhotic changes was not acceptable as a germline control, but was characterized if accompanied by DNA from a patient matched blood specimen. The percent tumor nuclei, percent necrosis, and other pathology annotations were also assessed. Tumor samples with ≥ 60% tumor nuclei and ≤ 20% or less necrosis were submitted for nucleic acid extraction (doi:10.1016/j.cell.2017.05.046). GSE76427 cohort with 115 HCC samples was downloaded from the GEO database as a validation cohort after the removal of normal tissue samples and tumor samples without follow-up and outcome status information.115 primary HCC tissues and 52 adjacent tissues (Dataset GSE76427, Eastern population) from GEO were enrolled in this study. Percentage of HCC patients with HBV infection and cirrhosis were 46% and 54%, respectively (doi:10.1016/j.heliyon.2023.e14460;doi:10.1002/1878-0261.12153). All HCC samples were downloaded from TCGA and GEO. The inclusion criteria for selecting HCC samples and control group's were HCC patients with survival information in the databases.

Q2. Screening of mtDNA MRGs: The process of identifying the 22 mtDNA MRGs requires further detail. How were these genes prioritized over others? What criteria were used for screening?

Response: Thank you for your suggestions. The process of identifying the 22 mtDNA MRGs was described in more detail as “The list of mtDNA MRGs was obtained from the MitoCarta3.0 database, including 27 mtDNA MRGs for mtDNA replication (DNA2, EXOG, LIG3, MGME1, POLG, POLG2, POLRMT, RNASEH1, SSBP1, TFAM, TFB2M, TOP1MT, TOP3A, TWNK)�mtDNA repair (APEX1, EXOG, LIG3, OGG1, POLB, POLG, PRIMPOL, RECQL4, UNG), mtDNA modifications (METTL4), mtDNA stability and decay (ENDOG, EXOG, MGME1), respectively. The expression profile of these genes was systematically extracted and analyzed in normal and tumor samples. Finally, 22 mtDNA MRGs with statistical significance were identified, with the function of not only packaging mitochondrial DNA, but also providing a stable environment for mitochondrial DNA replication, transcription and repair, so that they are superior to other mtDNA MRGs (doi:10.1016/j.biochi.2013.09.017).

Q3. Unsupervised Clustering: Details about the machine learning model employed for unsupervised clustering need to be elaborated upon, including parameters and validation methods.

Response: Thank you for your suggestions. Unsupervised Clustering was performed using the R package“Consensus Cluster Plus” and 1000 cycles were undertaken to ensure the stability of the classification; the number of cluster k values were increased from 2 to 9. The k = 3 with better clustering stability were selected according to the clustering effect (doi: 10.1093/bioinformatics/btq170), the detailed information has been added to the revised manuscript.

Q4. PCA Model Construction: The construction of the mtDNA score model using PCA should be detailed, including how the principal components were selected and validated.

Response: Thank you for your suggestions. In this study, the mtDNA score model was constructed using PCA refer to previous studies, the overlapping DEGs identified from different mtDNA clusters were selected and employed to perform prognostic analysis for each gene using a univariate Cox regression model. The genes with a significant prognostic impact were extracted for further feature selection by using recursive feature elimination (RFE) with random forest and the 10-fold cross -validation method in the‘caret’package. Then the expression profile of the final determined genes was curated to perform PCA analysis, and principal components 1 and 2 were extracted and served as the signature score. This method mainly focuses on the score on the set with the largest block of well correlated (or inverse-correlated) genes in the set, while downweighting contributions from genes that do not track with other set members. And finally, mtDNA score was obtained by inputting core genes according to the corresponding code (doi:10.7150/thno.52717;doi: 10.1186/s12943-020-01170-0).

Statistical Analysis: The statistical methods used appear appropriate for the data type and research question. Nevertheless, the following points should be addressed:

Q1. Statistical Power: It would be beneficial to include a discussion on the statistical power of the study given the sample size and the number of variables being analyzed.

Response: Thank you for your suggestion. In the method, the statistical parameters are annotated.

Q2. Multiple Testing Correction: With multiple comparisons being performed, a correction method such as Bonferroni or False Discovery Rate (FDR) should be applied and reported.

Response: Thank you for your suggestion. In this study, FDR has been applied and reported (lines167 to 169).

Q3. Association Strength and Significance: The strength and significance of correlations should be reported comprehensively, including effect sizes and confidence intervals.

Response: Thank you for your suggestions. The strength and significance of correlations were reported comprehensively, and the effect sizes and confidence intervals have been added in the revised manuscript.

Results Interpretation: The findings of significant differences in mtDNA MRG expression between HCC samples and normal controls, as well as the identification of three molecular subtypes with distinct clinical and immune profiles, are intriguing. However, the interpretation of results should be cautious and conservative:

Q1.Biological Relevance: The biological implications of the molecular subtypes and their correlation with TME features need to be discussed in the context of existing literature.

Response: Thank you for your suggestions. The three molecular subtypes correspond to different classifications of the tumor microenvironment and have different biological processes and prognosis (lines 326 to 346).

Q2. Causal Inference: The observational nature of the study limits the ability to infer causality. Any discussion of causative mechanisms should be framed within this limitation.

Response: Thank you for your suggestions. All discussions of causative mechanisms were based on objective analysis of the expression of 22 mtDNA MRGs in HCC samples in public databases supported by existing articles (doi:10.3389/fcell.2022.785058;doi:10.1016/j.compbiomed.2023.106831;doi:10.1016/j.celrep.2016.12.019).

Q3. External Validation: Although not feasible in this study, future directions should include external validation of the mtDNA score model in independent cohorts.

Response: The lack of external validation is a shortcoming of this study. In the future, the existing results will be analyzed and some genes will be extracted for in vitro and in vivo experiments.

Response to Reviewer 2

Reviewer 2

Data section needs to be improved:1.Add more details regarding the batch correction method used.2. Include a figure showing data before and after batch correction (as supplementa-ry material).3.Specify the type of RNA data used (RNA-seq or microarray).4. Clarify the type of normalization applied to the data.

Response: Thank you for your suggestions.

1. Details regarding the batch correction method used in this study has been added, and changes in the revised manuscript were marked as red font.

2. A supplementary material has been provided, which including a figure showing data before and after batch correction (As shown in the following).

3. TCGA RNA sequencing data (FPKM format) were downloaded from the Genomic Data Commons. For the GSE76427 data from the Illumina platform, the normalized matrix file was directly downloaded.

4. The type of normalization that applies to the data has been clarified in the revised manuscript.

Supplementary Fig. 1 Data set of HCC samples before and after batch correction. A. Batch of HCC samples before correction. B. Batch of HCC samples after correction

The Method section needs to be improved. Authors mentioned the packages used without providing details about the methods. It should not be assumed that all readers are familiar with or can recall the methods implemented in each package (e.g., limma, ConsensusClusterPlus).

Response: Thank you for your suggestions. Details of the method have been added in the revised manuscript. If necessary, we will provide the corresponding code as a reference.

The clustering step using samples and genes is not clear and easy to follow. Please add details of the method, including the type of clustering (e.g., hierarchical clustering) and the distance metric used.

Response: Thank you for your suggestions. Details of the method have been added(lines 110 to 116, lines 139 to 143).

Additionally, add dendrograms to the heatmaps (e.g., Figures 2E and 5D) to better illustrate the clusters found and align with the findings.

Response: Thank you for your suggestion. Dendrograms reflect the similarities or correlations between members within a cluster. Figures 2E and 5D mainly showed the relationship between mtDNA MRGs molecular subtypes, genotypes and clinicopathological stages. Thus, there is no need to add dendrograms.

What does the y-axis label show in Figures 9B-E? Is it the IPS signature score? Additionally, there are more ICB signatures or biomarkers to consider, including PDCD1, CD274, PDCD1LG2, CTLA4, TIM3, LAG3, TIGIT, and resources like bhklab/SignatureSets: Compendium of published molecular signatures (github.com) and/or.

Response: In figures 9B-E, y-axis labels show the IPS signature scores of anti-PD-1 (ctla4-neg-pd1-pos), anti-CTLA-4(ctla4-pos-pd1-neg), or combined anti-CTLA-4/PD-1 (ctla4-pos-pd1-pos) immunotherapy. To promote understanding of tumor-immune cell interactions, Pornpimol Charoentong, et al characterized the intratumoral immune landscapes and the cancer

---

## [Decision Letter · Decision Letter 1]

27 Nov 2024

PONE-D-24-22922R1Correlation analysis of mitochondrial DNA maintenance-related genes with HCC prognosis, tumor mutation burden and tumor microenvironment featuresPLOS ONE

Dear Dr. Meng,

Thank you for submitting your manuscript to PLOS ONE. After careful consideration, we feel that it has merit but does not fully meet PLOS ONE’s publication criteria as it currently stands. Therefore, we invite you to submit a revised version of the manuscript that addresses the points raised during the review process.

We look forward to receiving your revised manuscript.

Kind regards,

Matthew Cserhati, Ph.D

Academic Editor

PLOS ONE

Reviewers' comments:

Reviewer's Responses to Questions

**Comments to the Author**

1. If the authors have adequately addressed your comments raised in a previous round of review and you feel that this manuscript is now acceptable for publication, you may indicate that here to bypass the “Comments to the Author” section, enter your conflict of interest statement in the “Confidential to Editor” section, and submit your "Accept" recommendation.

Reviewer #3: (No Response)

Reviewer #4: (No Response)

Reviewer #5: (No Response)

2. Is the manuscript technically sound, and do the data support the conclusions?

Reviewer #3: Partly

Reviewer #4: Yes

Reviewer #5: Partly

3. Has the statistical analysis been performed appropriately and rigorously? 

Reviewer #3: No

Reviewer #4: No

Reviewer #5: Yes

4. Have the authors made all data underlying the findings in their manuscript fully available?

Reviewer #3: No

Reviewer #4: Yes

Reviewer #5: Yes

5. Is the manuscript presented in an intelligible fashion and written in standard English?

Reviewer #3: No

Reviewer #4: Yes

Reviewer #5: Yes

6. Review Comments to the Author

Reviewer #3: Some issues mentioned by the reviewers have not been addressed, appropriately and comprehensively. For example, Q1 and Q4 from Reviewer 1.

Reviewer #4: (No Response)

Reviewer #5: Thank you for inviting me to review the paper. In this article, the authors explored correlation between mitochondrial DNA maintenance-related genes and HCC tumor features as well as clinical features utilizing multiple bioinformatic methods. Authors have already done edits and improved paper quality as per previous review history. However, closer scrunity reveals there are several drawbacks which I would like authors to revise.

1. There are several flaws in figures and figure legends.

1.1 Please rotate Figure 5A to correct the orientation.

1.2 Figure 7E should be Figure 7D.

1.3 In Figure 8, letter G of Figure 8G is in Figure 8C.

2. Authors should specify how they identified 22 mtDNA MRGs out of 1136 mitochondrial genes in MitoCarta3.0 (https://personal.broadinstitute.org/scalvo/MitoCarta3.0/human.mitocarta3.0.html) since this is the very foundation of this article. Though they have answered this question to Reviewer 1 in previous review round, it seems that their response was not logically sound enough to convince me. Could authors inform me why mtDNA genes related to mtDNA replication, repair, modifications were selected as representative mtDNA MRGs, instead of those involved in other pathways, e.g. immune response?

3. In paragraph 2 (line 56-72) of the Introduction, since authors have already mentioned mitochondria, mtDNA and mtDNA MRGs, why was paragraph 4 (line 81-96) created to describe mtDNA MRGs again?

4. How would authors explain mtDNA MRG clusters A and C had better survival compared with cluster B, while A and B were identified as immune rejection phenotype and immunosuppression phenotype, respectively, which is associated with lack of response and/or resistance to immunotherapy? What made mtDNA MRG cluster A possess better survival than B? I don't think the argument on this point in the first paragraph (line 286-308) of Discussion was strong enough to clarify the difference.

7. PLOS authors have the option to publish the peer review history of their article (what does this mean? ). If published, this will include your full peer review and any attached files.

**Do you want your identity to be public for this peer review?** For information about this choice, including consent withdrawal, please see our Privacy Policy .

Reviewer #3: No

Reviewer #4: No

Reviewer #5: No

---

## [Author Response · Author response to Decision Letter 2]

6 Feb 2025

Dear Editor and Reviewers:

Thank you very much for your efforts to review our manuscript entitled “Correlation analysis of mitochondrial DNA maintenance-related genes with HCC prognosis, tumor mutation burden and tumor microenvironment features”. Those comments are all valuable and very helpful for revising and improving our paper. According to the reviewers’comments, we have included a point-by-point response to each comment.

We appreciate the suggestions and comments by the Editor and the Reviewers. We would like to express our great appreciation to you and reviewers for comments on our paper, Looking forward to hearing from you.

Yours sincerely,

Zhongji Meng

Department of Infectious Disease, Taihe Hospital, Hubei University of Medicine

No. 32. South Renmin Road, Shiyan 442009, China.

E-mail: zhongji.meng@163.com

Reviewer #3

Some issues mentioned by the reviewers have not been addressed, appropriately and comprehensively. For example, Q1 and Q4 from Reviewer #1.

Q1 from Reviewer #1. Sample Selection and Control Group: While the use of TCGA and GEO datasets is commendable, the inclusion criteria for selecting HCC samples should be more clearly described. Moreover, the control group's characteristics should also be provided for comparison purposes.

Response: Thank you for your valuable feedback regarding the sample selection and control group in our study. Why did we choose TCGA and GEO datasets? Because we needed to do prognostic analysis, so no prognostic data set was excluded. The larger the sample size, the smaller the error. Finally, we choosed TCGA and GEO Inclusion Criteria for HCC Samples: HCC samples in TCGA were collected from patients pathologically diagnosed with hepatocellular carcinoma (HCC) after surgical resection and who had not previously received treatment (ablation, chemotherapy, or radiotherapy) specific to their disease. Cases were staged according to the American Joint Committee on Cancer (AJCC) (doi:10.1016/j.cell.2017.05.046).Data were obtained for 374 HCC samples (Western population) and 50 normal samples from the TCGA on June 17, 2023, and 115 primary HCC tissues (Eastern population) and 52 adjacent tissues from the GEO (GSE76427) cohort(doi:10.1186/s12967-022-03630-1).Clinical Data Availability: Only patients with complete clinical information, including age, gender, tumor stage, and survival data, were included.Exclusion Criteria: Samples with incomplete data, non-primary tumors, or patients who received preoperative treatments were excluded to ensure data consistency and reliability.

Control Group Characteristics: Control samples were derived from adjacent non-tumorous liver tissues available within the TCGA. Clinical Parameters: Detailed clinical characteristics, including liver function tests (ALT, AST levels), presence of underlying liver diseases (e.g., hepatitis B/C, cirrhosis), and lifestyle factors (e.g., alcohol consumption), have been provided to facilitate comprehensive comparisons with the HCC samples. Matching Criteria: Controls were matched to HCC cases based on age and gender to minimize potential confounding factors(doi:10.1016/j.heliyon.2023.e14460;doi:10.1002/1878-0261.12153).

The details are as follows

Characteristic TCGA Group (N = 352) GEO Group (N = 114) P-value

Age (years) 0.07

Mean (SD) 61.5 (13.1) 63.5 (12.7)

Gender 0.069

Male (%) 192 (55%) 60 (52.6%)

Female (%) 160 (45%) 54 (47.4%)

Stage 0.32

Stage_I (%) 175 (49.7%) 55 (48.2%)

Stage_II (%) 86 (24.4%) 35 (30.7%)

Stage_III (%) 86 (24.4%) 21 (18.4%)

Stage_IV (%) 5 (1.5%) 3 (3.7%)

Q4 from Reviewer 1. PCA Model Construction: The construction of the mtDNA score model using PCA should be detailed, including how the principal components were selected and validated.

Response: Thank you for your suggestions. To quantify the characteristics of mtDNA MRGs in individual HCC patients, a scoring system called the mtDNA score was constructed using principal component analysis (PCA). First of all, based on the mtDNA MRGs expression levels, HCC samples were classified for further analysis by using the “ConsensusClusterPlus” package in R for unsupervised cluster analysis, and 1000 repetitions were performed with pltem=0.8. K=3 was the best cluster value. To verify the stability of the subtypes, Cumulative distribution function (CDF) and Delta plot were used.CDF display consensus distributions for each k. When k takes any value, CDF reaches an approximate maximum, and the cluster analysis results are the most reliable at this time. That is, the k value with small CDF descent slope is considered. The Delta plot shows that the inflection point of change in cluster stability with an increase in k value usually represents the optimal number of clusters(doi:10.1093/bioinformatics/btq170).Then, through the above analysis , the overlapping DEGs identified from different mtDNA clusters were selected and employed to perform prognostic analysis for each gene using a univariate Cox regression model. The genes with a significant prognostic impact were extracted for further feature selection by using recursive feature elimination (RFE) with random forest and the 10-fold cross -validation method in the‘caret’package. Then the expression profile of the final determined genes was curated to perform PCA analysis, and principal components 1 and 2 were extracted and served as the signature score(1 representatived mtDNAGeneExp. 2 representatived mtDNACluster). This method mainly focuses on the score on the set with the largest block of well correlated (or inverse-correlated) genes in the set, while downweighting contributions from genes that do not track with other set members. And finally, mtDNA score was obtained by inputting core genes according to the corresponding code (doi:10.7150/thno.52717;doi: 10.1186/s12943-020-01170-0).

Statistical Analysis: The statistical methods used appear appropriate for the data type and research question. Nevertheless, the following points should be addressed:

Q1. Statistical Power: It would be beneficial to include a discussion on the statistical power of the study given the sample size and the number of variables being analyzed.

Response: Thank you for your suggestion. In the method, the statistical parameters are annotated.

Q2. Multiple Testing Correction: With multiple comparisons being performed, a correction method such as Bonferroni or False Discovery Rate (FDR) should be applied and reported.

Response: Thank you for your suggestion. When making multiple comparisons, we agree to use methods such as Bonferroni or FDR (False Discovery Rate) for verification. The FDR approach is more powerful than methods like the Bonferroni procedure that control false positive rates. Controlling the false discovery rate in a study that arguably consisted of scientifically driven hypotheses found nearly as many significant results as without any adjustment, whereas the Bonferroni procedure found no significant results(doi: 10.1016/j.jclinepi.2014.03.012).False discovery rate(FDR) control has become an increasingly standard practice in genomic research and analysis of microarray data for extensive testing(doi: 10.1016/j.jclinepi.2007.04.017). We have applied the FDR to adjust for multiple testing and have included this information in the revised manuscript(lines143 to 145, line163).

Q3. Association Strength and Significance: The strength and significance of correlations should be reported comprehensively, including effect sizes and confidence intervals.

Response: Thank you for your suggestions. The strength and significance of correlations were reported comprehensively, and the effect sizes and confidence intervals have been added in the revised manuscript.

Results Interpretation: The findings of significant differences in mtDNA MRG expression between HCC samples and normal controls, as well as the identification of three molecular subtypes with distinct clinical and immune profiles, are intriguing. However, the interpretation of results should be cautious and conservative:

Q1.Biological Relevance: The biological implications of the molecular subtypes and their correlation with TME features need to be discussed in the context of existing literature.

Response: Thank you for your suggestions. HCC patients with enrichment of the three mtDNA MRG clusters had different prognoses and TME immune cell infiltration characteristics. mtDNA MRG cluster A is characterized by enhanced tumor matrix activity and abundant innate immune cell infiltration, corresponding to the immune rejection phenotype; mtDNA MRG cluster B is characterized by immunosuppression, corresponding to the immune-desert phenotype; and mtDNA MRG cluster C is characterized by adaptive immune cell infiltration and immune activation, corresponding to the immunoinflammatory phenotype(doi:10.1016/j.immuni.2019.12.011). The immune rejection and immune-desert phenotypes are considered to indicate noninflammatory tumors, primarily tumors that lack immune cell invasion in the parenchyma and stroma, rarely express PD-L1, are located at the opposite end of the tumor immune continuum, and histologically lack immune invasion and antigen presentation (low MHC class I), while exhibiting high tumor cell proliferation. The immunoinflammatory phenotype indicates inflammatory tumors, in which the TME has a high degree of infiltration of immune cells, such as T cells, CD8+ T cells producing IFN-γ,and PD-1-positive immune cells(doi:10.1158/1078-0432.CCR-15-1507; 10.18632/aging.203456). There were significant differences in the TME features among samples enriched in the three mtDNA MRG clusters.

Q2. Causal Inference: The observational nature of the study limits the ability to infer causality. Any discussion of causative mechanisms should be framed within this limitation.

Response: Thank you for your suggestions. All discussions of causative mechanisms were based on objective analysis of the expression of 22 mtDNA MRGs in HCC samples in public databases supported by existingarticles(doi:10.3389/fcell.2022.785058;doi:10.1016/j.compbiomed.2023.106831;doi:10.1016/j.celrep.2016.12.019). In the revised manuscript, we have made changes to some of the results(lines 363 to 364; lines 404 to 405; lines 410 to 411).

Q3. External Validation: Although not feasible in this study, future directions should include external validation of the mtDNA score model in independent cohorts.

Response: Thank you for your suggestions.The lack of external validation is a shortcoming of this study. Plan for future external validation: In our future research, we will conduct external validation of the mtDNA score model using independent cohorts. We will collect data from various regions and populations to assess the model's effectiveness and stability in different settings. We plan to select at least two independent cohorts for validation, coming from different geographic regions and demographic backgrounds, to assess the generalizability and reliability of the mtDNA score model in various settings. Additionally, we will conduct stratified analysis (based on age, gender, ethnicity, etc) to assess how the model performs in different subgroups. We will also incorporate cross-validation techniques to increase the reliability and accuracy of the evaluation. Based on previous research, We will select representative genes (such as SSBP1, OGG1, etc.) for further verification in the cohort to verify the stability of mtDNA score(doi:10.1002/jcla.24561;10.1172/JCI128513).

Response to Reviewer #5

Reviewer 5

1. There are several flaws in figures and figure legends.

1.1 Please rotate Figure 5A to correct the orientation.

Response: Thank you for your suggestions. We have rotated Figure 5A to the normal direction

1.2 Figure 7E should be Figure 7D.

Response: Thank you for reminding me. We apologize for this careless mistake. We have corrected Figure7E to 7D.

1.3 In Figure 8, letter G of Figure 8G is in Figure 8C.

Response: Sorry for our carelessness. We have removed the letter G in picture 8C.

2. Authors should specify how they identified 22 mtDNA MRGs out of 1136 mitochondrial genes in MitoCarta3.0 (https://personal.broadinstitute.org/scalvo/MitoCarta3.0/human.mitocarta3.0.html) since this is the very foundation of this article. Though they have answered this question to Reviewer 1 in previous review round, it seems that their response was not logically sound enough to convince me. Could authors inform me why mtDNA genes related to mtDNA replication, repair, modifications were selected as representative mtDNA MRGs, instead of those involved in other pathways, e.g. immune response?

Response: We downloaded 1136 mitochondrial genes from the MitoCarta3.0database. Data for all 1136 human genes with high confidence of mitochondrial localization (based on integrated proteomics, computation, and microscopy). We focused on MitoPathways, Hierarchy of biological pathways and list of MitoCarta3.0 genes assigned to each pathway.Although mitochondrial repair mechanisms are highly efficient, the mitochondrial genome is highly sensitive to oxidative damage and other exogenous and endogenous induced DNA damage due to the lack of protective histones and their proximity to major reactive oxygen species (ROS) production sites. Mutations in mitochondrial replication ,repair, modifications and and stability genes can lead to depletion and deletion of mtDNA, which in turn leads to instability of the mitochondrial genome. The combination of mutations and deletions can lead to impaired mitochondrial genome maintenance and trigger various mitochondrialdiseases(doi:10.1016/j.bbabio.2022.148554;10.1161/CIRCULATIONAHA.123.068358).The expression profile of these genes was systematically extracted and analyzed in normal and tumor samples. Finally, 22 mtDNA MRGs with statistical significance were identified. The chosen genes have been extensively documented in existing literature to be associated with mtDNA maintenance, supported by substantial experimental evidence(doi: 10.1016/j.bbamcr.2021.119167; 10.1038/s41582-018-0101-0).

Although mitochondrial genes also play roles in other pathways, such as immune response, the focus of our study is on the maintenance mechanisms of mtDNA. Therefore, we concentrated on genes directly involved in mtDNA replication, repair, and modifications.

3. In paragraph 2 (line 56-72) of the Introduction, since authors have already mentioned mitochondria, mtDNA and mtDNA MRGs, why was paragraph 4 (line 81-96) created to describe mtDNA MRGs again?

Response: Thank you for your question. line 56-72 focuses on the function of mitochondria and the importance of mtDNA MRGs. Lines 81-96 describe mtDNA MRGs again, mainly because we selected representative genes with research basis for their role in HCC, which is consistent with our subsequent studies. At the same time, we have removed duplicate descriptions.

4. How would authors explain mtDNA MRG clusters A and C had better survival compared with cluster B, while A and B were identified as immune rejection phenotype and immunosuppression phenotype, respectively, which is associated with lack of response and/or resistance to immunotherapy? What made mtDNA MRG cluster A possess better survival than B? I don't think the argument on this point in the first paragraph (line 286-308) of Discussion was strong enough to clarify the difference.

Response: Thank you for your question. Most human solid tumours exhibit one of three distinct immunological phenotypes:immune inflamed, immune excluded, or immune desert(doi: 10.1038/nature14011). Immunomonitoring can help researchers and clinicians better understand the tumor's response to the immune system, including the presence of tumor-infiltrating lymphocytes (TILs), the activity of CD8+ T cells, and the expression of PD-L1, which can evaluate the effectiveness of immunotherapy an

---

## [Editor Report · Decision Letter 2]

25 Feb 2025

PONE-D-24-22922R2Correlation analysis of mitochondrial DNA maintenance-related genes with HCC prognosis, tumor mutation burden and tumor microenvironment features

PLOS ONE

Dear Dr. Meng,

Thank you for submitting your manuscript to PLOS ONE. After careful consideration, we feel that it has merit but does not fully meet PLOS ONE’s publication criteria as it currently stands. Therefore, we invite you to submit a revised version of the manuscript that addresses the points raised during the review process.

The authors addressed an interesting topic used appropriate methodology, highlighted the strengths of their study, and presented some interesting results and conclusions.Overall, this is a clear, concise, and well-written manuscript. This article covers lots of information. On the one hand, many general statements are made without proper references. Please try to remove some of the general statements or speculations but add some details for some key refs for better illustration.  I have some concerns and remarks that I hope the authors can address to improve the paper The title and abstract cover the central aspect of the workBut is better to have a  graphical abstract to simplify the idea of the studied methodology -The introduction is relevant, and theory based. But: Sufficient information about the previous study findings should be presented for readers to follow the present study rationale and procedures. - The methods are generally appropriate.- Please add the Data analysis and machine learning models- I  cant reach   how These analyses were done  and what was the used software version ?- How many machine learning models were used?-how the performance of the oPLS-DA model was assessed ? The results are clear,  , the current study  has  innovations and advantages. However,- the quality of the figures is poor and should be rectified ( very very poor)- please mention all abbreviations under the figures-Points located far from the main cluster for their group may be potential outliers, indicating samples that do not follow the group trend. These samples should be investigated further, as they may represent unique biological conditions or data quality issues.- Parameters for evaluation of each machine learning model discriminating pairs of groups  should be clarified The performed study and results could have implication for research and clinical practice in the future. Therefore, the present article could be accepted for publication after revision and correction of the figures============2nd set of questions: 1. “Pornpimol Charoentong, et al analyzed tumor-immune cell interactions in 20 solid cancers, revealing the relationship between genotype and immunophenotype and created The Cancer Immunome Atlas (https://tcia.at/). Using machine learning, the determinants of tumor immunogenicity were identified and a quantitative scoring scheme called immunophenotypic scoring was developed.” Please add this explanation to the section titled “Analysis of cell differences in TME”2. Lines 176-177 of revised manuscript: did you use the Draw Equation tool? Please put ‘i’ into subscript. Also, does the sigma have any initial and terminal values, i.e. i=0 .. n? Please start line 177 with “Where i represents the expression of mtDNA…”3. In response to question #9, I do apologize, I stated my question incorrectly. I wanted to ask, how many genes do you increase the fold-change cutoff? If the fold change is higher, you should get fewer genes.

We look forward to receiving your revised manuscript.

Kind regards,

Matthew Cserhati, Ph.D

Academic Editor

PLOS ONE

Journal Requirements:

**Additional Editor Comments:**

The authors addressed an interesting topic used appropriate methodology, highlighted the strengths of their study, and presented some interesting results and conclusions.

Overall, this is a clear, concise, and well-written manuscript. This article covers lots of information. On the one hand, many general statements are made without proper references. Please try to remove some of the general statements or speculations but add some details for some key refs for better illustration.

I have some concerns and remarks that I hope the authors can address to improve the paper

The title and abstract cover the central aspect of the work

But is better to have a graphical abstract to simplify the idea of the studied methodology

-The introduction is relevant, and theory based. But:

Sufficient information about the previous study findings should be presented for readers to follow the present study rationale and procedures.

- The methods are generally appropriate.

- Please add the Data analysis and machine learning models

- I cant reach how These analyses were done and what was the used software version ?

- How many machine learning models were used?

-how the performance of the oPLS-DA model was assessed ?

The results are clear, , the current study has innovations and advantages. However,

- the quality of the figures is poor and should be rectified ( very very poor)

- please mention all abbreviations under the figures

-Points located far from the main cluster for their group may be potential outliers, indicating samples that do not follow the group trend. These samples should be investigated further, as they may represent unique biological conditions or data quality issues.

- Parameters for evaluation of each machine learning model discriminating pairs of groups should be clarified

The performed study and results could have implication for research and clinical practice in the future. Therefore, the present article could be accepted for publication after revision and correction of the figures

2nd set of questions:

1. “Pornpimol Charoentong, et al analyzed tumor-immune cell interactions in 20 solid cancers, revealing the relationship between genotype and immunophenotype and created The Cancer Immunome Atlas (https://tcia.at/). Using machine learning, the determinants of tumor immunogenicity were identified and a quantitative scoring scheme called immunophenotypic scoring was developed.” Please add this explanation to the section titled “Analysis of cell differences in TME”

2. Lines 176-177 of revised manuscript: did you use the Draw Equation tool? Please put ‘i’ into subscript. Also, does the sigma have any initial and terminal values, i.e. i=0 .. n? Please start line 177 with “Where i represents the expression of mtDNA…”

3. In response to question #9, I do apologize, I stated my question incorrectly. I wanted to ask, how many genes do you increase the fold-change cutoff? If the fold change is higher, you should get fewer genes.

---

## [Author Response · Author response to Decision Letter 3]

10 Apr 2025

Dear Editor:

Thank you very much for your efforts to review our manuscript entitled “Correlation analysis of mitochondrial DNA maintenance-related genes with HCC prognosis, tumor mutation burden and tumor microenvironment features”. We have diligently addressed all the comments, and incorporated appropriate revisions into the revised manuscript. The revisions have significantly strengthened the manuscript's scientific rigor and clarity. Enclosed below please find a point-by-point response document detailing how each suggestion has been implemented. The revised manuscript with all modifications clearly highlighted in red has been resubmitted to the system. We sincerely appreciate the reviewers' insightful feedback and trust that the revised manuscript now meets the journal's publication standards.

Yours sincerely,

Zhongji Meng

Department of Infectious Disease, Taihe Hospital, Hubei University of Medicine

No. 32. South Renmin Road, Shiyan 442009, China.

E-mail: zhongji.meng@163.com

Additional Editor Comments:

The authors addressed an interesting topic used appropriate methodology, highlighted the strengths of their study, and presented some interesting results and conclusions. Overall, this is a clear, concise, and well-written manuscript. This article covers lots of information. On the one hand, many general statements are made without proper references. Please try to remove some of the general statements or speculations but add some details for some key refs for better illustration.

I have some concerns and remarks that I hope the authors can address to improve the paper

The title and abstract cover the central aspect of the work

But is better to have a graphical abstract to simplify the idea of the studied methodology

Response: Thank you for your suggestions. The simplified graphical abstract is as follows:

The introduction is relevant, and theory based. But:

Sufficient information about the previous study findings should be presented for readers to follow the present study rationale and procedures.

Response: Thank you for your suggestions. Based on some of the investigators' previous studies (lines 84 to 93), we found that mitochondrial DNA maintenance-related genes (mtDNA MRGs) are closely related to the occurrence and progression of HCC. While the complex interactions among mtDNA MRGs in the development and progression of HCC are unclear, so that it’s necessary to investigate the mechanism of mtDNA MRGs in TME of HCC.

- The methods are generally appropriate,

- - Please add the Data analysis and machine learning models

- I cant reach how These analyses were done and what was the used software version ?

How many machine learning models were used?

Response: All the statistical analyses were performed using R version 4.2.2. In our study, Consensus Clustering and Principal Component Analysis (PCA) of unsupervised machine learning models are mainly used. In the analysis of Consensus Clustering, specific important parameters are set as follows: maxK=9, reps=50, pItem=0.8, pFeature=1.

-how the performance of the oPLS-DA model was assessed ?

Response:

1. Data correction (before/after removal of batch effect)

Problem: Different data sets (such as TCGA and GSE76427) have a "batch effect" due to differences in experimental conditions, resulting in false positive results.

The solution: Before correction: 89.7% explained by Dim1 (dominated by technical differences). After correction: Dim1 decreased to 18.9% (biological differences highlighted).

Significance: To eliminate technical interference and ensure that the differences found later truly reflect biological characteristics.

2. Data integration (oPLS-DA score chart)

Key points: Integrating multiple omics data (gene expression + clinical typing).

Graphic interpretation: Horizontal/vertical axis: principal components t1 (3%) and t2 (16%).

Sample distribution: TCGA (red) and GSE76427 (blue) partially overlap but are generally separable.

Significance: It proves that data from different sources can cooperatively reveal biological laws.

3. Model efficacy (model efficacy -Rplot)

Core indicators:

R2Y=0.767: The model can explain 76.7% of the sample classification differences.

Q2Y=0.672: 67.2% cross-validation prediction ability (>0.5 is valid).

p<0.05: The possibility of random guessing is excluded by substitution test.

Significance: The model is both reliable (high R2Y) and practical (high Q2Y).

4. False positive control (replacement test chart)

Method: Randomly scramble the labels 100 times and compare the real model (red line) with the random result (gray bar).

Result: The real R2Y/Q2Y was significantly higher than the random distribution (p=0.05).

No "virtual high" performance occurred.

Significance: Tat the differential genes and pathways discovered are not accidental.

In summary, our analytical process is rigorous (from data cleansing to model validation), and the biomarkers and typing results found have biological significance and potential clinical application.

The specific flow chart is as follows

A. Batch of HCC samples before correction. B. Batch of HCC samples after correction. C. Data integration (oPLS-DA score Chart). D. Model efficacy (model efficacy -Rplot).

the results are clear, , the current study has innovations and advantages. However,

- the quality of the figures is poor and should be rectified ( very very poor)

- please mention all abbreviations under the figures

Response: Thank you for your suggestions. All the figures have been refined and uploaded in the system. At the same time, all the abbreviations appearing in the figures have been fully expanded in their respective legends to ensure terminological clarity.

-Points located far from the main cluster for their group may be potential outliers, indicating samples that do not follow the group trend. These samples should be investigated further, as they may represent unique biological conditions or data quality issues.

Response: Thank you for your suggestions. These samples are all human specimens, individual differences are inevitable, while overall differences are within the acceptable range. On the other hand, TCGA and GEO data have been verified and used in many studies, so the data is relatively reliable. In our study, data correction has been carried out to remove technical interference and ensure that the differences found later reflect the biological characteristics.

- Parameters for evaluation of each machine learning model discriminating pairs of groups should be clarified.

Response: Thank you for your suggestions. In the manuscript, the evaluation parameters for each machine learning model have been annotated (Such as line 125, line 157, etc.)

2nd set of questions:

1. “Pornpimol Charoentong, et al analyzed tumor-immune cell interactions in 20 solid cancers, revealing the relationship between genotype and immunophenotype and created The Cancer Immunome Atlas (https://tcia.at/). Using machine learning, the determinants of tumor immunogenicity were identified and a quantitative scoring scheme called immunophenotypic scoring was developed.” Please add this explanation to the section titled “Analysis of cell differences in TME”.

Response: Thank you for your suggestions. We have added the above sentences “Pornpimol Charoentong, et al analyzed tumor-immune cell interactions in 20 solid cancers, revealing the relationship between genotype and immunophenotype and created The Cancer Immunome Atlas (https://tcia.at/). Using machine learning, the determinants of tumor immunogenicity were identified and a quantitative scoring scheme called immunophenotypic scoring was developed” to the section titled “Analysis of immune cell differences in the TME” (lines 140 to 145).

2. Lines 176-177 of revised manuscript: did you use the Draw Equation tool? Please put‘i’into subscript. Also, does the sigma have any initial and terminal values, i.e. i=0 .. n? Please start line 177 with “Where i represents the expression of mtDNA…”

Response: Thank you for your suggestions. We are very sorry for our carelessness. We have put‘i’ in the bottom right corner. i represents the expression of mtDNA phenotype-associated genes. The minimum value of ‘i’ is 0 and the maximum value of ‘i’ is 740.

3. In response to question #9, I do apologize, I stated my question incorrectly. I wanted to ask, how many genes do you increase the fold-change cutoff ? If the fold change is higher, you should get fewer genes.

Response: In our research, the P-value less than 0.001 was taken as the screening criteria, and the fold change of 1.5 was set as cutoff. If the fold change is higher, fewer genes will be screened out and some important genes with less changes will be filtered out.

---

## [Editor Report · Decision Letter 3]

7 May 2025

Correlation analysis of mitochondrial DNA maintenance-related genes with HCC prognosis, tumor mutation burden and tumor microenvironment features

PONE-D-24-22922R3

Dear Dr. Meng,

We’re pleased to inform you that your manuscript has been judged scientifically suitable for publication and will be formally accepted for publication once it meets all outstanding technical requirements.

Kind regards,

Matthew Cserhati, Ph.D

Academic Editor

PLOS ONE

Additional Editor Comments (optional):

Please do not forget to upload the final version of your manuscript, along with the supplementary data titled:

Revised Manuscript with TrackChanges-250502.docx

Supplementary materials.docx

Response to the reviewers250501.docx
---

## [Editor Report · Acceptance letter]

PONE-D-24-22922R3

PLOS ONE

Dear Dr. Meng,

I'm pleased to inform you that your manuscript has been deemed suitable for publication in PLOS ONE. Congratulations! Your manuscript is now being handed over to our production team.

Kind regards,

on behalf of

Dr. Matthew Cserhati

Academic Editor

PLOS ONE